# Self-Paced Contrastive Learning for Semi-supervised Medical Image Segmentation with Meta-labels

**Jizong Peng**[*]
ETS Montreal
jizong.peng.1@etsmtl.net

**Ping Wang**
ETS Montreal
ping.wang.1@ens.etsmtl.ca

**Christian Desrosiers**
ETS Montreal
christian.desrosiers@etsmtl.ca

**Marco Pedersoli**
ETS Montreal
marco.pedersoli@etsmtl.ca

## Abstract

The contrastive pre-training of a recognition model on a large dataset of unlabeled data often boosts the model's performance on downstream tasks like image classification. However, in domains such as medical imaging, collecting unlabeled data can be challenging and expensive. In this work, we consider the task of medical image segmentation and adapt contrastive learning with meta-label annotations to scenarios where no additional unlabeled data is available. Meta-labels, such as the location of a 2D slice in a 3D MRI scan, often come for free during the acquisition process. We use these meta-labels to pre-train the image encoder, as well as in a semi-supervised learning step that leverages a reduced set of annotated data. A self-paced learning strategy exploiting the weak annotations is proposed to further help the learning process and discriminate useful labels from noise. Results on five medical image segmentation datasets show that our approach: *i*) highly boosts the performance of a model trained on a few scans, *ii*) outperforms previous contrastive and semi-supervised approaches, and *iii*) reaches close to the performance of a model trained on the full data.

## 1 Introduction

Since the emergence of deep learning [26], there has been an active debate on the importance of pre-training neural networks. Precursor works [17] showed that pre-training a convolutional neural network with an unsupervised task (e.g., denoising autoencoders [49]) could lead to a better performance in the final supervised task. As the amount of labeled training data increased, thanks to large datasets like ImageNet [13], it was however found that pre-training could actually hinder performance [38]. This makes sense in light of recent studies showing, for instance, that symmetries in large networks induce many equivalent local minima [16, 35, 47] in which a pre-trained model can get stuck. Recently, contrastive learning has renewed the interest in unsupervised pre-training [37]. Several works [8, 10, 11, 20, 62] have found that pre-training a model with a contrastive loss can improve its performance on a subsequent supervised training task, often outperforming a network with supervised pre-training on ImageNet. While this has reopened the debate on the benefit of pre-training, it offers little help for domains where data is scarce such as medical imaging. In medical imaging, not only are labels expensive since they come from highly-trained experts like radiologists, but images are also hard to obtain due to the need for costly equipment (e.g., MRI or CT scanner) and privacy regulations.

Over the last years, a breadth of semi-supervised learning approaches have been proposed for medical image segmentation, including methods based on attention [32], adversarial learning [60],

---

[*]Corresponding author

35th Conference on Neural Information Processing Systems (NeurIPS 2021).

temporal ensembling [12, 55], co-training [40, 63], data augmentation [6, 61] and transformation consistency [5]. The common principle of these approaches is to add an unsupervised regularization loss using unlabeled images, which is optimized jointly with a standard supervised loss on a limited set of labeled images. Despite reducing significantly the amount of labeled data required for training, current semi-supervised learning methods still suffer from important drawbacks which impede their use in various applications. Thus, a large number of unlabeled images is often necessary to properly learn the regularization prior. As mentioned before, this may be impossible in medical imaging scenarios where data is hard to obtain. Moreover, these methods also need a sufficient amount of labeled data, otherwise the learning may collapse [36].

In a recent work, Chaitanya et al. [7] showed that unsupervised pre-training can be useful to learn a segmentation task with very few samples, by leveraging the meta information of medical images (e.g., the position of a 2D image in the 3D volume). While achieving impressive accuracy with as few as two volumes, this work has significant limitations. First, it relies on the strong assumption that the global or local representations of 2D images are similar if their locations within the volume or feature map are related. This assumption does not always hold in practice since volumes may not be well aligned, or due to the high variability of structures to segment. Second, it requires dividing the 2D images of a 3D volume in an arbitrary number of hard partitions that are contrasted, while the structure to segment typically varies gradually within the volume. Third, they do not exploit the full range of available meta data, for instance the patient ID or cycle phase of cardiac cine MRI, nor evaluate the benefit of combining several types of meta information in pre-training. Last, their approach leverages meta data only in pre-training, however this information could further boost performance if used while learning the final segmentation task, in a semi-supervised setting.

Our work addresses the limitations of current semi-supervised and self-supervised approaches for segmentation by proposing a novel self-paced contrastive learning method, which takes into account the noisiness of weak labels from meta data and exploits this data jointly with labeled images in a semi-supervised setting. The detailed contributions of this paper are as follows:

– We propose, to our knowledge, the first self-paced strategy for contrastive learning which dynamically adapts the importance of individual samples in the contrastive loss. This helps the model deal with noisy weak labels that arise, for instance, from misaligned images or splitting a 3D volume in arbitrary partitions.
– We demonstrate the usefulness of a contrastive loss on meta-data for improving the performance of a final task, not only in pre-training but also as an additional loss in semi-supervised training.
– We show that combining multiple meta-labels in our self-paced contrastive learning framework can improve performance on the final task, compared to using them independently. Our results also demonstrate the benefit of combining contrastive learning with temporal ensembling to further boost performance.

We empirically validate our contributions on five well-known medical imaging datasets, and show the proposed approach to outperform the contrastive learning method of [7] as well as several state-of-the-art semi-supervised learning methods for segmentation [41, 43, 50, 58, 60]. In the results, our approach obtains a performance close to fully supervised training with very few training scans.

## 2 Related work

We focus our presentation of previous works on two machine learning sub-fields that are most related to our current work: self-supervision, which includes contrastive learning, and self-paced learning.

**Self-supervision and contrastive learning**  Self-supervision is a form of unsupervised learning where a pretext task is used to pre-train a model so to better perform a downstream task. Examples of pretext tasks are learning to sort a sequence [28, 54], predicting rotations [19, 18], solving a jigsaw puzzle [33] and many others [14, 15, 25, 46, 59]. Most of these methods can improve performance on the downstream task when labeled data for training is scarce. However, when a large and general-enough dataset of labeled images like ImageNet [13] is available, a simple supervised pre-training may sometimes provide better results [22, 34]. Recently, unsupervised contrastive learning [8, 20, 10] was shown to boost performance even when learning a downstream task on a large dataset, and to improve over a model pre-trained in a supervised manner on a large dataset.

This approach is based on the simple idea of enforcing similarity in the representation of two examples from the same class (*positive* pairs), and increase representation dissimilarity on pairs from different classes (*negative* pairs) [37, 48]. In image analysis tasks, positive pairs are typically defined as two

transformed versions of the same image, for instance using a geometric or color transformation, while negative ones are any other pair of images [8, 10, 11, 20].

In a recent work, Khosla et al. [24] showed that, when combined with true semantic labels, a contrastive learning based "pre-train and fine-tune" pipeline performed surprisingly well, outperforming conventional training with cross-entropy in some cases. The work in [62] extended this idea to pixel-level tasks like semantic segmentation, clustering the representations of pixels in an image according to their labels. Applying a similar strategy to medical image segmentation, Chaitanya et al. [7] leveraged meta-labels from 3D scans in a local and global contrastive learning framework to improve performance when training with limited data. However, positive and negative pairs in their contrastive loss are defined using noisy "weak" labels, arising for example from misaligned images or an arbitrary partitioning of 3D volumes, which may lead to learning sub-optimal representations. The work in [56] mitigated this problem by imposing a maximum distance along the $z$ axis between slices forming positive pairs. Moreover, existing approaches only exploit meta-labels in pre-training, instead of considering them jointly with labeled images in a semi-supervised strategy. Our work extends these approaches with a self-paced learning method that adapts the importance of positive pairs dynamically during training, and focuses the learning on the most reliable ones. In contrast to [7], we also exploit contrastive learning as regularization loss in semi-supervised training and show that further improvements can be achieved when combining it with a temporal ensembling strategy like Mean Teacher [12, 55]. A recent approach by Chen et al. [10] also performs contrastive learning for image recognition in a semi-supervised setting. In this approach, a large teacher model is trained with a unsupervised contrastive loss and then fine-tuned with a small fraction of labeled data. A student model is trained afterwards using a knowledge distillation technique. Hence, unlike our method, there is no joint optimization of the supervised and contrastive objectives. In this work, we show that significant improvements can be achieved by jointly optimizing these two objectives.

**Self-paced learning** A sub-category of curriculum learning (CL) [2], self-paced learning (SPL) is inspired by the learning process of humans that gradually incorporates easy to hard samples in training [27]. The effectiveness of such strategy has been validated in various computer vision tasks [52, 23, 57]. Jiang et al. [23] proposed an SPL method considering both the difficulty and diversity of training examples, which outperformed conventional SPL methods that ignore diversity. Zhang et al. [57] incorporated SPL in a DNN fine-tuning process for object detection, to cope with data ambiguity and guide the learning in complex scenarios. The usefulness of SPL when training with a limited budget or when the training data is corrupted by noise was also studied in recent work [53]. So far, the application of SPL to image segmentation remains limited. Wang et al. [52] presented an SPL method for lung nodule segmentation, where the uncertainty of each sample prediction in the loss is controlled by the SPL regularizer. Similarly, a self-paced co-training method was proposed in [51] for the semi-supervised segmentation of medical images. Related to our work, Wang et al. [31] proposed a margin preserving contrastive learning framework for domain adaptation that uses a self-paced strategy in self-training. Compared to this approach, which uses SPL *outside* the contrastive loss to select confident pseudo-labels for self-training, our method incorporates it *within* the loss via importance weights that are learned jointly with network parameters.

## 3 Proposed method

In this section, we present our self-paced contrastive learning approach for segmentation that leverages intrinsic meta information extracted from medical volumetric images. Our method effectively pre-trains a segmentation model with a self-paced variant of contrastive learning that is more robust to noisy annotations. The same approach is also used to further boost the segmentation accuracy in a semi-supervised setting, where only very limited pixel-wised annotations are used. In the following subsections, we detail the formulation of the proposed approach.

### 3.1 Contrastive learning with meta-labels

Given a batch of $N$ images from a dataset $\mathcal{D}_u$ of unlabeled images, unsupervised contrastive learning approaches [8, 21, 37] aim at finding a feature extractor $f(\cdot)$ that gives similar representations for two augmented instances of the same image and different ones for those of two separate images, regardless of their true classes. This can be achieved by creating an augmented set of samples indexed by $i \in I \equiv 1, \ldots, 2N$, with two augmented samples for each original image in the batch. The

following loss is then optimized:

$$\mathcal{L}_{\text{unsupCon}} = \frac{1}{2N} \sum_{i=1}^{2N} - \log \frac{\exp\left(\mathbf{z}_i^\mathsf{T} \mathbf{z}_{j(i)}/\tau\right)}{\sum_{a \in \mathcal{A}(i)} \exp\left(\mathbf{z}_i^\mathsf{T} \mathbf{z}_a/\tau\right)} \tag{1}$$

In this loss, $\mathbf{z}_i = \frac{f(\mathbf{x}_i)}{\|f(\mathbf{x}_i)\|}^2$ is the $L_2$-normalized representation of an image $\mathbf{x}_i$ in the augmented batch (i.e., the *anchor*) and $j(i)$ is the index of the other augmented sample from the same image (i.e., the *positive*). $\mathcal{A}(i) \equiv I \setminus \{i\}$ contains all indexes of the augmented set except $i$, and has a size of $2N-1$. Finally, $\tau$ is a small temperature factor that helps gradient descent optimization by smoothing the landscape of the loss.

This approach works well when a large dataset of unlabeled images is available. However, the number of available images is small in our case. To alleviate this problem, our contrastive learning framework also leverages meta-labels arising from the structure of the data. Following [7], we consider the 2D slices of a given set of $M$ volumetric scans as our training data, and extract various meta-labels for each 2D image (e.g., patient ID, position of the slice in the volume, etc). More generally, we suppose that each image $\mathbf{x}_i$ has set of $K$ meta-labels denoted as $y_i^k \in \{1, \ldots, C_k\}$, where $C_k$ is the number of class labels for meta information $k \in \{1, \ldots, K\}$. The contrastive loss for the meta-label $k$ is then defined as

$$\mathcal{L}_{\text{con}}^k = \frac{1}{2N} \sum_{i=1}^{2N} \frac{1}{|\mathcal{P}^k(i)|} \sum_{j \in \mathcal{P}^k(i)} \underbrace{- \log \frac{\exp\left(\mathbf{z}_i^\mathsf{T} \mathbf{z}_j/\tau\right)}{\sum_{a \in \mathcal{A}(i)} \exp\left(\mathbf{z}_i^\mathsf{T} \mathbf{z}_a/\tau\right)}}_{\ell_{ij}} \tag{2}$$

where $\mathcal{P}^k(i) = \{j \in I \mid y_j^k = y_i^k\} \cup \{j(i)\}$ are the indexes of augmented samples with same label as $\mathbf{x}_i$, or coming from the same original image. By minimizing this loss, the feature extractor learns to group together representations with the same class and push away those from different ones.

In medical image segmentation, encoder-decoder based networks such as U-Net [45] and its variants are widely utilized thanks to their symmetric design and appealing performance on various dataset. Such network $F(\cdot)$ decomposes in two parts, an encoder $E(\cdot)$ that summarizes the global context of an input 2D slice into a low-dimensional representation, and a decoder $D(\cdot)$ that takes as input the representation and gradually recovers its spatial resolution using side information such as skip connections or pooling indexes. Previous work on contrastive learning showed that pre-training both the encoder and decoder separately helped the downstream segmentation task [7]. In preliminary experiments, we found that pre-training the decoder gave marginal improvements and thus focused our method on the encoder. Specifically, we consider the features of the encoder as a single vector $E(\mathbf{x}) \in \mathbb{R}^d$ and use a shallow non-linear projector $g(\cdot)$ called *head* to obtain the final normalized embedding $\mathbf{z}_i = \frac{g(E(\mathbf{x}_i))}{\|g(E(\mathbf{x}_i))\|}$.

## 3.2 Self-paced learning to mitigate noisy meta-labels

The supervised contrastive loss of Equ. (2) can actually hurt the learning of representations in pre-training if the positive pairs are obtained with "weak" or noisy labels. For instance, if using patient ID as meta-label, we will force the encoder to cluster the representations of all 2D slices in a 3D volume, *including* those containing mainly background noise. Likewise, grouping together the slices in the same partition of two volumes hinders pre-training if the volumes are not fully aligned and/or their partitions cover different regions of the structure to segment.

To overcome this problem, we propose a self-paced strategy for contrastive learning which assigns an importance weight $w_{ij} \in [0, 1]$ to the specific loss of each positive pair $(i, j)$, defined as $l_{ij}$ in Equ. (2). A self-paced regularization term $R_\gamma(w_{ij})$, controlled by the learning pace parameter $\gamma$, is added to give a greater importance (i.e., larger $w_{ij}$) to pairs that are more confident (i.e., smaller $\ell_{ij}$), and vice-versa. The learning pace $\gamma$ is increased over training so that high-confidence pairs are considered in the beginning and then less-confident ones are gradually added as training progresses. We achieve this by defining the following self-paced contrastive loss optimized over both encoder parameters and importance weights:

$$\mathcal{L}_{\text{SP-con}}^k = \frac{1}{2N} \sum_{i=1}^{2N} \frac{1}{|\mathcal{P}^k(i)|} \sum_{j \in \mathcal{P}^k(i)} w_{ij}\, \ell_{ij} + R_\gamma(w_{ij}) \tag{3}$$

---

[2]We omit the nonlinear projector head for the sake of simplification.

Following standard SPL approaches [23], we define the regularizer $R_\gamma$ such that the weights are monotone decreasing with respect to the loss $l_{ij}$ (i.e., harder examples are given less importance) and monotone increasing with respect to the learning pace (i.e., a larger $\gamma$ increases the weights). In this work, we consider two regularizer functions, based on hard thresholding and linear imputation:

$$R_\gamma^{\text{hard}}(w_{ij}) = -\gamma\, w_{ij}; \qquad R_\gamma^{\text{linear}}(w_{ij}) = \gamma\big(\tfrac{1}{2}w_{ij}^2 - w_{ij}\big). \tag{4}$$

**Optimization process**   We minimize the loss in Equ. (3) by optimizing alternatively with respect to the encoder parameters $\theta_E$ or importance weights $w_{ij}$, while keeping the other fixed. With fixed $w_{ij}$, we update $\theta_E$ via stochastic gradient descent where the gradient is given by:

$$\nabla_{\theta_E} \mathcal{L}_{\text{SP-con}}^k = \frac{1}{2N} \sum_{i=1}^{2N} \frac{1}{|\mathcal{P}^k(i)|} \sum_{j\in\mathcal{P}^k(i)} w_{ij}\, \nabla_{\theta_E} \ell_{ij}. \tag{5}$$

As can be seen, the gradient of low-confidence pairs $(i,j)$ will be scaled down by their weight $w_{ij}$, thus these pairs will contribute less to the learning. Then, given a fixed $\theta_E$ we compute the optimal weights $w_{ij}^*$ by solving the following problem:

$$w_{ij}^* = \underset{w_{ij}\in[0,1]}{\arg\min}\ w_{ij}\,\ell_{ij} + R_\gamma(w_{ij}) \tag{6}$$

The following proposition gives the optimal solution for the hard and linear SPL regularization strategies.

**Proposition 1.** *Given the definitions of $R_\gamma^{\text{hard}}$ and $R_\gamma^{\text{linear}}$ in Equ. (4), the closed-form solutions to Equ. (6) are given by*

$$w_{ij}^{\text{hard}} = \begin{cases} 1, & \text{if } \ell_{ij} \le \gamma \\ 0, & \text{else} \end{cases}; \qquad w_{ij}^{\text{linear}} = \max\Big(1 - \frac{1}{\gamma}\ell_{ij},\, 0\Big). \tag{7}$$

*Proof.* For the hard regularizer $R_\gamma^{\text{hard}}$ the problem becomes

$$\min_{w_{ij}\in[0,1]}\ w_{ij}\,\ell_{ij} - \gamma\, w_{ij} = (\ell_{ij} - \gamma)w_{ij} \tag{8}$$

If $\ell_{ij} - \gamma \ge 0$, since we are minimizing, the optimum is obviously $w_{ij} = 0$. Else, if $\ell_{ij} - \gamma < 0$, the minimum is achieved for $w_{ij} = 1$. Combining these two results gives the hard threshold of Equ. (7).

A similar approach is used for the linear regularizer $R_\gamma^{\text{hard}}$. In this case, the problem to solve is

$$\min_{w_{ij}\in[0,1]}\ w_{ij}\,\ell_{ij} + \gamma\big(\tfrac{1}{2}w_{ij}^2 - w_{ij}\big) = \tfrac{\gamma}{2}w_{ij}^2 + (\ell_{ij} - \gamma)\, w_{ij} \tag{9}$$

If $l_{ij} \ge \gamma$, since $w_{ij} \ge 0$, the minimum is reached for $w_{ij} = 0$. Else, if $\ell_{ij} < \gamma$, we find the optimum by deriving the function w.r.t. $w_{ij}$ and setting the result to zero, giving

$$w_{ij} = 1 - \frac{1}{\gamma}\ell_{ij}. \tag{10}$$

Since both $\gamma$ and $\ell_{ij}$ are non-negative, we have that $w_{ij} \in [0,1]$, hence it is a valid solution. Considering both cases simultaneously, we therefore get the linear rule of Equ. (7). $\qquad\square$

These update rules in Equ. (7) can be explained intuitively. For a given $\gamma$, the hard threshold rule only considers confident pairs with $\ell_{ij} \le \gamma$ and ignores the others. In contrast, the linear rule weighs each pair proportionally to $\gamma$ and the inverse of $\ell_{ij}$, emphasizing more confident ones.

**Selecting the learning pace parameter**   One of the main challenges in self-paced learning methods is selecting the learning pace parameter $\gamma$. If $\gamma$ is too small, all pairs will be ignored and there will be no learning. Conversely, if $\gamma$ is too large, all pairs will be considered regardless of their confidence, which corresponds to having no self-paced learning. The following proposition provides insights on how to set this parameter during training.

**Proposition 2.** *The loss $\ell_{ij}$ related to a given pair $(i,j)$ in the SPL objective of Equ. (3) is bounded by $\log 2(N-1) - 2/\tau \le \ell_{ij} \le \log 2N + 2/\tau$, where $N$ is the batch size.*

*Proof.* We start by rewriting $l_{ij}$ equivalently as

$$\ell_{ij} = \log \frac{\sum_{a \in \mathcal{A}(i)} \exp\left(\mathbf{z}_i^\mathsf{T} \mathbf{z}_a / \tau\right)}{\exp\left(\mathbf{z}_i^\mathsf{T} \mathbf{z}_j / \tau\right)} = \log\left(1 + \sum_{a \in \mathcal{A}(i) \setminus j} \frac{\exp\left(\mathbf{z}_i^\mathsf{T} \mathbf{z}_a / \tau\right)}{\exp\left(\mathbf{z}_i^\mathsf{T} \mathbf{z}_j / \tau\right)}\right) \quad (11)$$

Since the representation vectors $\mathbf{z}_i$ are $L_2$-normalized, their dot product is a cosine similarity falling in the range $[-1, 1]$. To minimize $l_{ij}$, we then need to minimize the dot product in the numerator inside the sum and maximize the one in the denominator. Using $|\mathcal{A}(i) \setminus j| = 2N - 2$, we get

$$\ell_{ij}^{\min} = \log\left(1 + (2N-2)\frac{e^{-1/\tau}}{e^{1/\tau}}\right) = \log\left(1 + 2(N-1)e^{-2/\tau}\right) \geq \log 2(N-1) - 2/\tau. \quad (12)$$

Similarly, we maximize $\ell_{ij}$ by doing the opposite:

$$\ell_{ij}^{\max} = \log\left(1 + (2N-2)\frac{e^{1/\tau}}{e^{-1/\tau}}\right) = \log\left(1 + 2(N-1)e^{2/\tau}\right) \leq \log 2N + 2/\tau. \quad (13)$$

$\square$

This proposition tell us that using $\gamma = \ell_{ij}^{\max}$ guarantees that all pairs are used in the loss, for both the hard and the linear SPL regularizers. Additionally, when using the hard regularizer $R_\gamma^{\mathrm{hard}}$, $\gamma = \ell_{ij}^{\min}$ is the minimum learning pace so that at least one pair can be selected.

**The complete SPL loss**   To exploit the information in all available meta-labels, our final loss combines the contrastive losses $\mathcal{L}_{\mathrm{SP\text{-}con}}^k$ for meta-labels $k = 1, \ldots, K$:

$$\mathcal{L}_{\mathrm{SP\text{-}con}} = \sum_{k=1}^K \lambda_k \, \mathcal{L}_{\mathrm{SP\text{-}con}}^k \quad (14)$$

Here, $\lambda_k \geq 0$ is a coefficient controlling the relative importance of the $k^{th}$ meta-label in the final loss, which is determined by grid search on a separate validation set.

### 3.3   Semi-supervised segmentation with contrastive learning

In previous work [7], contrastive learning has mostly been used for pre-training the model. Here, we show that it can further boost results in a semi-supervised setting, where training is performed with a limited set of samples. In this setting, in addition to the unlabeled images $\mathcal{D}_u$, a small amount of pixelwise-annotated images $\mathcal{D}_l$ are also available. To incorporate the knowledge from meta information in a semi-supervised setting, we modify our self-paced contrastive loss as

$$\mathcal{L}_{\text{semi-sup}} = \mathcal{L}_{\text{sup}} + \lambda_{\text{reg}} \mathcal{L}_{\text{reg}} + \lambda_{\text{SP}} \mathcal{L}_{\mathrm{SP\text{-}con}}, \quad (15)$$

where $\mathcal{L}_{\text{sup}}$ is the loss computed on labeled data (cross-entropy loss in our work), $\mathcal{L}_{\text{reg}}$ is the regularization loss normally used in semi-supervised approaches (in our experiments we use Mean Teacher) and $\mathcal{L}_{\mathrm{SP\text{-}con}}$ is our self-paced contrastive loss. Last, $\lambda_{\text{reg}}$ and $\lambda_{\text{sp}}$ are weights balancing the different loss terms which are determined by grid search.

## 4   Experimental setup

To assess the performance of the proposed self-paced contrastive learning, we carry out extensive experiments on five benchmark datasets with different experimental settings. In this section, we briefly describe these datasets and give implementation details for our method. For further information, the reader can refer to the Supplementary Material.

### 4.1   Datasets

Five clinically-relevant benchmark datasets for medical image segmentation are used for our experiments: the Automated Cardiac Diagnosis Challenge (ACDC) dataset [3], the Prostate MR Image Segmentation 2012 Challenge (PROMISE12) dataset [29], and Multi-Modality Whole Heart Segmentation Challenge (MMWHS) dataset [64], as well as the Hippocampus and Spleen segmentation

datasets from [1]. These datasets contain different anatomic structures and present different acquisition resolutions. For the contrastive loss, we exploit meta-labels on slice position and patient identity. Additionally, for ACDC, we consider the cardiac phase (i.e., systole or diastole) as a third source of meta-data. For all datasets, we split images into training, validation and test sets, which remain unchanged during all experiments. We train the model with only a few scans of the dataset as labeled data (the rest of the data is used without annotations as in a semi-supervised setting) and report results in terms of 3D DSC metric [4] on the test set. Details on the training set split, data pre-processing, augmentation methods and evaluation metrics can be found in the Supplementary Material.

For all datasets, we report the segmentation performance by varying the number of labeled scans across experiments. For the ACDC dataset, this number ranges from 1 to 4, representing 0.5% to 2% of all available data. For PROMISE12, we use 3 to 7 scans, representing 6% to 14% of the whole data. For MMWHS, we use 1 and 2 annotated scans, corresponding to 10% and 20% of the training data. We use 1 to 4 scans as annotated data for the Hippocampus dataset, representing 0.5% to 0.2% of the whole data, and 2 to 4 scans for the Spleen dataset, which corresponds to 5.7% to 11.4% of the whole available training data. Note that once randomly selected, those labeled volumes are fixed across the different experiments. Selecting labeled scans per experiment yielded significant variances (up to 11.25% in term of 3D DSC), as shown in the Supplementary Material. We include the segmentation results for both Hippocampus and Spleen datasets in the Supplementary Material.

## 4.2 Network architecture and optimization parameters

We use `PyTorch` [39] as our training framework and, following [7], employ the U-Net architecture [45] as our segmentation network. This 2D-based networks often works well for data with anisotropic acquisition resolutions. Moreover, it has a lower computational cost and require less GPU memory than its 3D counterparts. Network parameters are optimized using stochastic gradient descent (SGD) with a RAdam optimizer [30]. We provide the detailed training hyper-parameters in the Suppl. Material. For the pre-training process, we obtain representations by projecting the encoder's output to a vector of size 256, using a simple MLP network with one hidden layer and LeaklyReLU activation function, following [9]. Our proposed self-paced contrastive learning objective, defined in Equ. (3), involves a learning pace parameter $\gamma$ set as

$$\gamma \;=\; \gamma_{\mathrm{start}} \,+\, (\gamma_{\mathrm{end}} - \gamma_{\mathrm{start}}) \times \left( \frac{\texttt{cur\_epoch}}{\texttt{max\_epoch}} \right)^{p} \tag{16}$$

where $\gamma_{\mathrm{start}}, \gamma_{\mathrm{end}}$ are hyper-parameters controlling the importance weights in the beginning and the end of training, and `cur_epoch`, `max_epoch` are the current training epoch and the total number of training epochs respectively. $p$ controls how fast $\gamma$ increases during the optimization procedure.

## 5 Results

In this section, we first compare the hard and linear regularization strategy for SPL on ACDC. Then, we evaluate all components of our method in a comprehensive ablation study with a reduced set of training data on the different datasets. Finally, we compare our method with the most promising approaches for semantic segmentation in medical imaging, with reduced training data.

## 5.1 Hard vs. linear self-paced regularization

Table 1 reports the validation 3D DSC score for the hard and linear SPL, while training with different $p$ in Equ. (16), and different numbers of annotated scans on the ACDC dataset. We observe that both SPL strategies ($R_\gamma^{\mathrm{hard}}$ and $R_\gamma^{\mathrm{linear}}$) effectively help improve performance, however the linear strategy always leads to a higher improvement. This is because the hard strategy only employs binary weights, i.e., $w_{ij} \in \{0, 1\}$, whereas the linear strategy gradually increases $w_{ij}$ and therefore provides a smoother optimization.

In Fig. 1 (a), we plot the value of $\gamma$ over epochs for different values of $p$, and show in (b) the corresponding expectation of $w_{ij}$ for all positive pairs. We observe that, for a large $p$, $\gamma$ tends to be small for most of the training and mainly increases in the very end of the process, resulting in small $w_{ij}$ for positive pairs. In contrast, when $p = 1/2$, we see a rapid increase of weights $w_{ij}$ during training, which results in higher segmentation scores. This observation is inline with the findings from [44] and [42] on different tasks, where raising rapidly the self-paced learning rate in the first

Figure 1: Self-paced strategy for $\gamma$. (a) Evolution of $\gamma$; (b) Expectation of $w_{ij}$ over training epochs.

Table 1: 3D DSC Performance on ACDC for hard and linear SP strategy and different values of $p$.

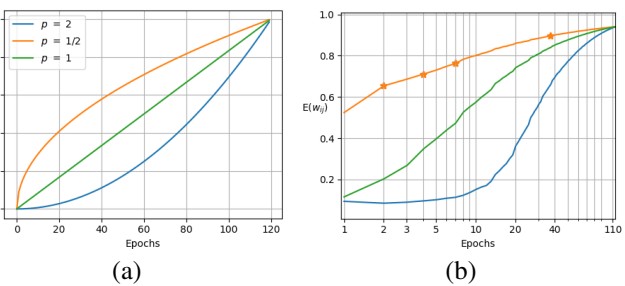

(a)                              (b)

| SP type | $p$ | ACDC | | |
|---|---|---|---|---|
| | | 1 scan | 2 scans | 4 scans |
| Baseline | | 57.53% | 67.06% | 75.64% |
| Linear | $^1/_2$ | **74.40%** | **80.34%** | **81.86%** |
| | 1 | 72.06% | 79.54% | 81.03% |
| | 2 | 59.72% | 70.36% | 80.05% |
| Hard | $^1/_2$ | 64.42% | 78.26% | 80.07% |
| | 1 | **72.01%** | **79.80%** | **80.24%** |
| | 2 | 71.86% | 72.14% | 76.19% |

half of the training benefits the generalization performance. In the Suppl. Material, we also present a concrete evaluation of $w_{ij}$ for three different scans during model optimization, corresponding to the four orange star markers in Fig. 1 (b). Since we found that $R_\gamma^{\text{linear}}$ strategy works better than $R_\gamma^{\text{hard}}$, we will use $R_\gamma^{\text{linear}}$ for all following experiments.

## 5.2 Ablation study

Table 2 summarizes the 3D DSC performance on test set for three datasets (ACDC, PROMISE12 and MMWHS) with very limited labeled data. At the top of the table, we report the number of labeled scans used and, for every result, also give in parenthesis the standard deviation computed with 3 different random seeds for parameter initialization. In the second and third columns of the table, we provide the loss used for the pre-training, if any, and the loss for the downstream training.

**Upper and lower bounds** We present results for a *Baseline* which uses only the annotated scans with cross-entropy as standard supervised loss $\mathcal{L}_{\text{sup}}$, and for *Full Supervision* where the same loss is used with all available data and associated annotations (175 for ACDC, 40 for Prostate and 10 for MMWHS). These two rows represent lower and upper bounds on the expected performance of the different variants of our approach.

**Unsupervised contrastive loss** We evaluate the performance of pre-training the network encoder with *Unsupervised Contrastive* loss as in [8], where two augmented versions of the same image are considered as a positive pair. In all datasets, this loss improves over our baseline model, although the improvement is limited because the amount of unlabeled data available is still reduced compared to the settings of previous work on unsupervised contrastive learning [8, 20, 11, 10, 62]. We also add our self-paced learning strategy on top of this contrastive loss, and call this modified model *Unsupervised Contrastive + SP*. As meta-labels may be noisy, performance is increased in almost all experiments, especially when fewer labels are available.

**Pre-training contrastive loss on meta-data** We report the performance of a model pre-trained with a *Contrastive* loss on meta-labels. The meta-labels are 3D slice location $\mathcal{L}_{\text{con}}^1$, patient identity $\mathcal{L}_{\text{con}}^2$ and cardiac phase $\mathcal{L}_{\text{con}}^3$ (only for ACDC). We find that that slice position always gives the highest accuracy among all meta-labels, and largely outperforms the unsupervised contrastive loss. While all meta-labels increase performance compared to unsupervised contrastive loss, their combination leads to the best results in most cases.

**Pre-training self-paced contrastive loss on meta-data** Next, we evaluate the model pre-trained with a Self-Paced Contrastive loss on meta-labels (*SP-Con (pre-train)*). As with the unsupervised contrastive loss, the self-paced approach also successfully improves the segmentation quality compared to treating all positive pairs equally.

**Semi-supervised** We report the performance of a model without any pre-training, but using the unlabeled data during training with our proposed self-paced contrastive loss (*SP-Con (semi-sup)*). It can be seen that performance is inferior to Contrastive pre-training on meta-data, however the improvement is still quite relevant and, in most cases, superior to unsupervised pre-training.

**Pre-trained and semi-supervised** Our next subsection reports results for the combination of Self-Paced Contrastive learning used for both pre-training and semi-supervised training (*SP-Con*

Table 2: 3D DSC performance (and standard deviation) for different components and approaches on three medical image datasets with a few labelled scans.

| Method | Pretrain | Train | ACDC | | | PROMISE12 | | | MMWHS | |
|---|---|---|---|---|---|---|---|---|---|---|
| | | | 1 scan | 2 scans | 4 scans | 3 scans | 5 scans | 7 scans | 1 scan | 2 scans |
| Baseline | – | $\mathcal{L}_{\mathrm{sup}}$ | 57.53 (1.18) | 67.06 (0.68) | 75.64 (0.15) | 35.02 (2.59) | 59.03 (1.25) | 72.81 (1.08) | 70.94 (0.84) | 80.25 (0.38) |
| Full Supervision (*all labels*) | – | $\mathcal{L}_{\mathrm{sup}}$ | 88.06 (0.20) | | | 89.70 (0.51) | | | 88.27 (1.23) | |
| Unsup. Con. | $\mathcal{L}_{\mathrm{con}}^{\mathrm{unsup.}}$ | $\mathcal{L}_{\mathrm{sup}}$ | 65.14 (2.53) | 72.88 (3.00) | 76.56 (1.34) | 38.74 (8.47) | 60.99 (5.20) | 75.28 (1.72) | 74.12 (1.45) | 80.57 (2.50) |
| Unsup. Con. + SP | $\mathcal{L}_{\mathrm{SP}}^{\mathrm{unsup.}}$ | $\mathcal{L}_{\mathrm{sup}}$ | 67.35 (1.98) | 75.11 (0.92) | 76.87 (0.84) | 40.21 (6.63) | 67.37 (0.99) | 75.14 (0.50) | 74.30 (1.81) | 80.71 (0.83) |
| Contrastive | $\mathcal{L}_{\mathrm{con}}^1$ $\mathcal{L}_{\mathrm{con}}^2$ $\mathcal{L}_{\mathrm{con}}^3$ | $\mathcal{L}_{\mathrm{sup}}$ | 70.57 (0.96) 63.63 (1.80) 64.52 (1.34) | 78.59 (0.79) 73.30 (1.25) 76.81 (0.97) | 79.60 (0.49) 76.83 (0.91) 77.66 (0.56) | 57.44 (4.89) 55.50 (3.83) – | 75.21 (1.94) 69.95 (1.06) – | 80.02 (1.28) 78.93 (0.63) – | 76.53 (1.79) 74.71 (0.29) – | 83.05 (2.68) 82.41 (0.33) – |
| SP-Con (*pre-train*) | $\mathcal{L}_{\mathrm{SP}}^1$ $\mathcal{L}_{\mathrm{SP}}^2$ $\mathcal{L}_{\mathrm{SP}}^3$ | $\mathcal{L}_{\mathrm{sup}}$ | 73.99 (1.27) 69.26 (1.69) 65.18 (1.50) | 81.01 (1.44) 76.34 (0.60) 79.05 (.026) | 82.83 (0.26) 78.34 (0.42) 81.04 (0.16) | 58.81 (2.35) 56.80 (1.59) – | 75.28 (1.49) 69.75 (0.47) – | 80.71 (1.27) 79.02 (0.18) – | 77.20 (0.87) 76.67 (0.48) – | 82.87 (0.39) 83.10 (1.51) – |
| SP-Con (*semi-sup*) | – | $\mathcal{L}_{\mathrm{sup}}+$ $\mathcal{L}_{\mathrm{SP}}^1$ $\mathcal{L}_{\mathrm{SP}}^2$ $\mathcal{L}_{\mathrm{SP}}^3$ | 67.34 (0.74) 60.82 (0.98) 62.52 (0.46) | 73.74 (0.51) 68.06 (1.09) 68.39 (0.26) | 77.27 (0.12) 77.10 (0.35) 77.24 (0.17) | 54.50 (1.53) 41.67 (1.59) – | 70.49 (1.33) 61.04 (1.46) – | 76.95 (0.81) 75.98 (0.98) – | 73.82 (0.68) 73.43 (1.33) – | 81.63 (0.39) 78.08 (1.88) – |
| SP-Con (*both*) | $\mathcal{L}_{\mathrm{SP}}^1$ $\mathcal{L}_{\mathrm{SP}}^2$ $\mathcal{L}_{\mathrm{SP}}^3$ | $\mathcal{L}_{\mathrm{sup}}+$ $\mathcal{L}_{\mathrm{SP}}^1$ $\mathcal{L}_{\mathrm{SP}}^2$ $\mathcal{L}_{\mathrm{SP}}^3$ | 75.66 (1.94) 70.47 (0.93) 70.08 (0.96) | 80.37 (0.36) 76.58 (0.45) 78.70 (0.51) | 82.35 (0.58) 78.37 (0.22) 80.19 (0.28) | 68.79 (2.63) 56.68 (2.64) – | 77.38 (1.90) 72.21 (1.32) – | 80.55 (0.75) 77.28 (1.86) – | 76.58 (1.00) 75.33 (0.62) – | 82.69 (0.39) 82.39 (0.27) – |
| SP-Con (*both*) + Mean Teacher | $\mathcal{L}_{\mathrm{SP}}^1$ $\mathcal{L}_{\mathrm{SP}}^2$ $\mathcal{L}_{\mathrm{SP}}^3$ $\mathcal{L}_{\mathrm{SP}}^{1-3}$ | $\mathcal{L}_{\mathrm{sup}}$ $\mathcal{L}_{\mathrm{MT}}+$ $\mathcal{L}_{\mathrm{SP}}^1$ $\mathcal{L}_{\mathrm{SP}}^2$ $\mathcal{L}_{\mathrm{SP}}^3$ $\mathcal{L}_{\mathrm{SP}}^{1-3}$ | 78.76 (0.26) 75.30 (0.68) 73.94 (0.54) **79.80** (0.33) | 82.14 (0.19) 79.67 (0.26) 81.29 (0.09) **83.20** (0.25) | 84.42 (0.18) 82.65 (0.32) 83.21 (0.05) **84.84** (0.15) | 74.06 (1.13) 61.39 (1.33) – **74.47** (0.36) | 82.50 (0.91) 77.74 (1.07) – **83.78** (0.30) | 84.14 (0.35) 83.92 (0.39) – **84.52** (0.17) | 78.82 (0.34) 75.94 (0.90) – **78.97** (0.52) | 84.90 (0.58) 84.57 (0.61) – **84.87** (0.11) |

*(both)*). Although the loss is the same, its use during pre-training and as additional regularization in a semi-supervised setting brings additional improvements.

**Pre-trained and semi-supervised with Mean Teacher**  We then evaluate the model using both pre-training and semi-supervised (as the previous setting) but with an additional Mean-Teacher for semi-supervision (*SP-Con (both) + Mean-Teacher*). By combining our approach with a simple Mean-Teacher method, our results on all datasets are further boosted, approaching the performance of fully supervised training but using a very low number of annotated scans. This is the model that is used in the comparison with the state-of-the-art.

## 5.3  Comparison with the state-of-the-art

Table 3: 3D DSC performance (and standard deviation) of our method and other approaches on three medical image datasets with few labelled scans. Bold red-colored values are the best performing methods, underlined blue-colored ones correspond to the second best performing method.

| Method | ACDC | | | PROMISE12 | | | MMWHS | |
|---|---|---|---|---|---|---|---|---|
| | 1 scan | 2 scans | 4 scans | 3 scans | 5 scans | 7 scans | 1 scan | 2 scans |
| Entropy Min. [50] | 60.47 (1.03) | 69.81 (0.99) | 76.19 (1.21) | 53.47 (5.70) | 65.66 (0.42) | 73.52 (2.71) | 72.28 (0.58) | 78.39 (1.54) |
| Mix-up [58] | 60.87 (1.28) | 67.45 (1.04) | 76.18 (0.49) | 41.38 (2.80) | 64.55 (1.93) | 73.56 (0.61) | 71.50 (0.54) | 80.12 (0.84) |
| Adv. Training [60] | 63.05 (0.80) | 70.68 (0.27) | 75.89 (0.94) | 61.58 (2.10) | 71.00 (1.20) | 81.05 (1.34) | 73.47 (1.42) | 80.40 (0.93) |
| Mean Teacher [43] | 62.85 (0.67) | 72.84 (0.22) | 79.12 (0.08) | 52.96 (1.97) | 68.38 (2.04) | 77.37 (0.87) | 72.36 (1.35) | 81.01 (0.57) |
| Discrete MI [41] | 69.27 (1.41) | 77.74 (0.42) | 80.06 (0.24) | 47.77 (3.58) | 68.29 (2.35) | 77.63 (1.13) | 72.38 (1.04) | 82.45 (1.36) |
| Contrastive [7] | 70.05 (2.66) | 79.11 (2.02) | 81.25 (2.15) | 61.15 (2.95) | 74.62 (1.69) | 80.08 (1.39) | 76.45 (0.62) | 82.93 (0.42) |
| Our Method | **79.80** (0.33) | **83.20** (0.25) | **84.84** (0.15) | **74.47** (0.36) | **83.78** (0.30) | **84.52** (0.17) | **78.97** (0.52) | **84.87** (0.11) |

We compare our method with other approaches that aim to improve training with few annotated images/scans. Table 3 presents results in terms of 3D DSC score for approaches based on data augmentation [58], pre-training the weights on both encoder and decoder of the model [7] and various semi-supervised learning methods [50, 60, 43, 41]. As with the ablation study, we report results for the ACDC, PROMISE12 and MMWHS datasets. A detailed explanation of the experimental setup of each method and results for the other two datasets can be found in the Supplementary Material.

To have a fair comparison, for all methods, we used grid search on the validation set to tune the hyper-parameters. For most methods, the improvement with respect to the baseline trained with only the supervised loss is quite limited and varies depending on the dataset and the number of annotated scans used. For instance, *Adversarial training* performs quite well on the PROMISE12 dataset (scans in this dataset exhibit more variability in terms of intensity contrast), but not so well

on ACDC. Likewise, *Mean-Teacher* does not perform well for 1 or 2 annotated scans in ACDC but, when increasing the scans to 4, it outperforms most of the other methods. The global and local *Contrastive* loss using meta-data manages to obtain an excellent improvement on all datasets. However, our approach still yields substantial improvements with respect to that method. This is due to our proposed self-paced learning strategy, as well as the combined use of the contrastive loss for pre-training and semi-supervised learning.

## 6  Discussion and conclusion

In this paper, we proposed a technique based on contrastive loss with meta-labels that can highly improve the performance of a medical image segmentation model when training data is scarce. It was shown that, with a reduced amount of unlabeled images, unsupervised contrastive loss is not very effective. Instead, in the context of medical images, additional meta-data is freely available and, if properly used, can greatly boost performance. We presented results on five well-known medical image datasets and have shown that the accuracy of the contrastive loss with meta-labels can be boosted by the use of self-paced learning. Our self-paced contrastive learning method can be used during pre-training as well as a regularization loss during semi-supervised training, and the combination of the two can further boosts results. Finally, we have compared our approach with the state-of-the-art in semi-supervised learning, and have shown that the simple combination of our approach with multiple meta-data and a simple semi-supervised approach as Mean Teacher is more effective than previous approaches. While using a few scans, our method can approach fully supervised training.

**Social impact and limitations**  The proposed method can have an effective and practical impact in terms of medical imaging analysis in hospitals and health centers. As shown in our experiments, it produces an accurate medical image segmentation with a very reduced set of annotated data. This has the potential of helping radiologists and other clinicians using medical images, which can in turn contribute to a better diagnosis and reduced costs. While our empirical evaluation has shown excellent results with very limited data, using fewer annotated images also increases chances of over-fitting potential outliers in the data that may lead to erroneous or misleading results. A further study on the reliability of medical image segmentation with reduced images is therefore recommended.

## 7  Acknowledgment

We acknowledge the support of the Natural Sciences and Engineering Research Council of Canada (NSERC Grant No. RGPIN-2018- 04825) and Fonds de recherche du Québec – Nature et technologies (FRQNT Grant No. B2X 276565). This research was also enabled in part by support provided by Calcul Québec (www.calculquebec.ca) and Compute Canada (www.computecanada.ca).

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
