*Supplementary Material:*
# Self-Paced Contrastive Learning for Semi-supervised Medical Image Segmentation with Meta-labels

**Jizong Peng**
ETS Montreal
`jizong.peng.1@etsmtl.net`

**Ping Wang**
ETS Montreal
`ping.wang.1@ens.etsmtl.ca`

**Christian Desrosiers**
ETS Montreal
`christian.desrosiers@etsmtl.ca`

**Marco Pedersoli**
ETS Montreal
`marco.pedersoli@etsmtl.ca`

## 1 Conceptual diagram and algorithm flow of proposed method

We illustrate the principle of our proposed self-paced contrastive learning mechanism in Fig. 1, where slice position is used as the meta-label. Our loss is computed on top of the conventional contrastive loss, while considering the self-paced coefficient $w_{ij}$ given a batch of unlabeled images. The self-paced coefficient $w_{ij}$, which measures thee reliability of a positive pair $(i, j)$, is calculated as in Equ. (6). We also include in **Algorithm 1** the algorithm flow of our method used with arbitrary meta-labels.

Figure 1: Conceptual diagram for our proposed self-paced contrastive learning method with meta-labels.

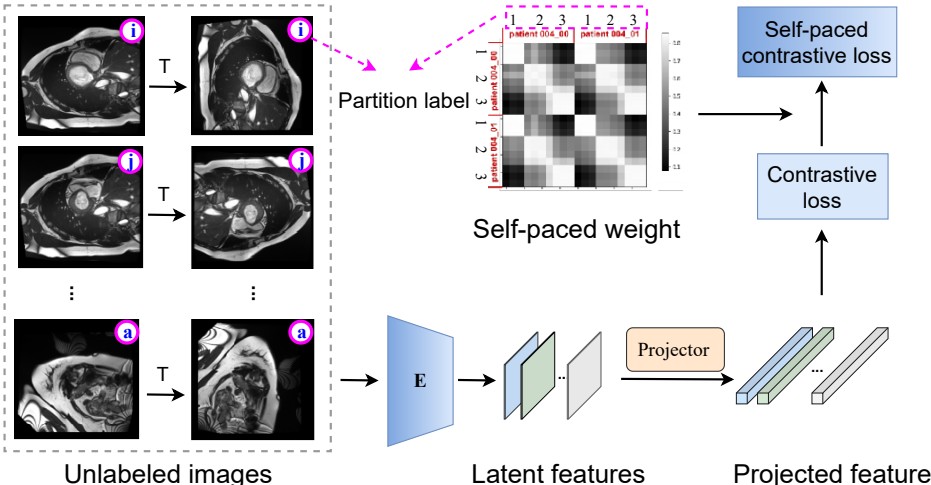

## 2 Dataset description

- **ACDC dataset** This publicly-available dataset [2] contains 200 short-axis cine-MRI scans obtained from 100 patients. Scans were acquired using 1.5 and 3 T systems with an in-plane resolution from $0.70 \times 0.70$ mm to $1.92 \times 1.92$ mm and a through-plane resolution from 5 mm to 10 mm. Volumetric images were obtained for end-diastolic (ED) (100 scans)

---

**Algorithm 1:** Self-paced contrastive learning in pretraining stage.

---

**Input:** Unlabeled dataset $\mathcal{U}$ and their respective meta-label set $D_{\text{meta}}$; Encoder of the segmentation
network $E(\cdot)$; Temperature $\tau$; Learning pace $\gamma$ scheduler ;

**Output:** Pre-trained model parameters $\{\boldsymbol{\theta}\}$ for $E(\cdot)$;

---

Initialize network parameters $\boldsymbol{\theta}$;

Initialize hyper-parameters: learning pace: $\gamma \leftarrow \gamma_0 = \text{Scheduler}\,(0)$ ;

**for** epoch $= 1, \dots, n_{\text{epochs}}$ **do**

    **for** n $= 1, \dots, n_{\text{iter}}$ **do**

        Sample unlabeled training batch $\{\mathcal{U}_n\}$;

        For all $\mathbf{x}_u \in \mathcal{U}_n$, do random transformation and get $\mathbf{x}_u^T$;

        Compute $z$ using non-linear project $g(\cdot)$ for the features $E(\mathbf{x}_u)$;

        Compute the sample-wise contrastive loss using Eq. (2):

$$\ell_{ij} = -\log \frac{\exp\left(z_i^{\intercal} z_j / \tau\right)}{\sum_{a \in \mathcal{A}(i)} \exp\left(z_i^{\intercal} z_a / \tau\right)};$$

        Compute self-paced importance weight $\omega_{ij}$ using Eq. (6):

$$w_{ij}^{*} = \underset{w_{ij} \in [0,1]}{\arg\min}\; w_{ij}\,\ell_{ij} + R_{\gamma}(w_{ij});$$

        Compute self-paced contrastive loss using Eq. (3):

$$\mathcal{L}_{\text{SP-con}}^{k} = \frac{1}{2N} \sum_{i=1}^{2N} \frac{1}{|\mathcal{P}^k(i)|} \sum_{j \in \mathcal{P}^k(i)} w_{ij}\,\ell_{ij} + R_{\gamma}(w_{ij});$$

        According to Eq. (14), do a batch gradient descent step on the model' parameters $\boldsymbol{\theta}$;

        Update the model' parameters $\boldsymbol{\theta}$;

    Adjust the SGD learning rate;

    Update learning pace according to the scheduler: $\gamma \leftarrow \text{Scheduler}\,(\text{epoch})$

**return** $\{\boldsymbol{\theta}\}$ ;

---

and end-systolic (ES) (100 scans) phases of the cardiac cycle. Ground-truth segmentation masks are provided for the following four regions of interest: left ventricle endocardium (LV), left ventricle myocardium (Myo), right ventricle endocardium (RV), and background. Due to a highly-variable resolution, we slice 3D scans through-plane into 2D images, and re-sample these 2D images to a fixed resolution of $1.0 \times 1.0$ mm. For each scan, intensities were normalized using the $1\%$ and $99\%$ percentile of the scan's intensity histogram before performing slicing. Normalized 2D images are then cropped to a size of $384 \times 384$. For our experiments, we used a random split of 175 scans as our training set, from which we randomly select 1, 2 or 4 scans as our labeled data, and considered others as unlabeled images[1]. We then divided the remaining 25 scans into a validation set consisting of 8 scans and a test set with 17 scans. Both the validation and test sets were set aside during model optimization. We employed a diverse set of data augmentations during training, for both labeled and unlabeled images, which include random crops of $224 \times 224$ pixels, random flips, random rotations within $[-45, 45]$ degrees, and color jitters.

We leveraged the rich meta information available in ACDC to obtain three types of meta-labels. Following [3], we first considered slice position and patient identity as meta-labels in our experiments, referring respectively as $\mathcal{L}_{\text{con/SP}}^1$ and $\mathcal{L}_{\text{con/SP}}^2$ the standard contrastive loss and self-paced contrastive loss based on these meta-labels. As [3], we defined the position of a 2D image in a volume based on the partition of this volume into $S = 3$ equal-sized groups of consecutive slices. Additionally, we used the cardiac phase of the scan (i.e., ED or ES) as third meta-label, and write as $\mathcal{L}_{\text{con/SP}}^3$ the contrastive losses using this meta-label. We detail the metal labels in the next section.

– **PROMISE12 dataset**   The second dataset [7] used to evaluate our method focuses on prostate MRI segmentation. It comprises multi-centric transversal T2-weighted MR images from 50 subjects, acquired with scanners from multiple vendors and different scanning protocols. Image resolution ranges from $15 \times 256 \times 256$ to $54 \times 512 \times 512$ voxels with a spacing between $2 \times 0.27 \times 0.27$ and $4 \times 0.75 \times 0.75$ mm$^3$. We sliced scans into 2D images along the short-axis and resized these images to a size of $256 \times 256$ pixels. Intensities were once again normalized based on the $1\%$ and $99\%$ percentiles of the intensity histogram for each scan before the slicing operation. We randomly selected 40 scans as our training set, and used 3, 5 or 7 scans from this set as our labeled data. We considered a validation

---

[1]Once selected, we fixed these splits across all experiments.

set with 4 scans and a test set with 6 scans. For data augmentation, we utilized the same set of transformations as with the ACDC dataset, except that we limit random rotations to $[-10,10]$ degrees. Similar to [3], we adopted slice position ($S = 5$ partitions) and patient identity as meta-labels for this dataset.

– **MMWHS dataset**  The third dataset considered in our evaluation, the Multi-Modality Whole Heart Segmentation (MMWHS) dataset [17], consists of high resolution CT images from 20 subjects. Four segmentation classes were used in our experiments: left ventricle myocardium (LVM), left atrium blood cavity (LAC), left ventricle blood cavity (LVC) and ascending aorta (AA). Following a similar protocol as with the ACDC dataset, volumetric images were first normalized based on their intensity histogram, then sliced along the short-axis, and finally resized to a resolution of $256 \times 256$ pixels. We randomly selected 10 scans as our training set, from which 1 or 2 were used as our labeled data and the others as unlabeled data. Validation and test sets contain 4 and 6 scans, respectively. The same set of transformation as with the ACDC dataset was used to augment images on the fly during training. Once again, we adopted slice position ($S = 7$ partitions) and patient identity as meta-labels for this dataset.

– **Hippocampus dataset**  The fourth dataset, as a sub-track of Medical Segmentation Decathlon [1], aims to segment hippocampus from 260 T1-sequence MRI images acquired from both healthy adults and adults with a non-affective psychotic disorder. As before, volumetric images were normalized according to their histogram and sliced to 2D images along the short axis with a spatial size of $96 \times 96$ pixels. We randomly split the images into training, validation and test set, consisting of 223, 12 and 25 scans, respectively. To perform semi-supervised segmentation, we then chose 1, 2, or 4 scans from these training examples as the labeled ones, while keeping others as unlabeled. We used the same data augmentation as the ACDC dataset and considered slice position ($S = 3$ partitions) as the meta-label.

– **Spleen dataset**  The last dataset [1] consists of patients undergoing chemotherapy treatment for liver metastases. A total of 41 portal venous phase CT scans were included in the dataset with acquisition and reconstruction parameters described in [1]. Similar to the previous datasets, 2D slices were obtained by slicing the high-resolution CT volumes along the axial plane. Intensities in each slice were clipped to a range of $[-100, 400]$ and resulting images resized to a resolution of $512 \times 512$ pixels. We randomly split the dataset into training, validation and test sets, comprising CT scans of 35, 2, and 5 patients respectively. To evaluate algorithms in a semi-supervised setting, we then randomly chose 2 or 4 scans from the training set as labeled examples and considered remaining images as unlabeled. We again applied the same set of data augmentations as for the ACDC dataset and employed slice position ($S = 5$ partitions) as the meta-label for this dataset.

## 3  Meta information visualization

In Fig. 2-4, we visualize examples of images from the three first datasets, corresponding to different patients (ACDC, PROMISE12 and MMWHS), slice partition (ACDC, PROMISE12 and MMWHS), and phase of the cardiac cycle (ACDC only). Although a given slice partition exhibits a high-level structural similarity across different volumes, we also observe important variability in corresponding images, due to differences in acquisition conditions, individual anatomy of subjects and imperfect image registration. Thus, without human interaction, this meta-label can lead to the learning of noisy representations. The second meta-label, patient identity, reflects global differences between scans. These differences are particularly notable for the PROMISE12 and MMWHS datasets, where the structural shape and the image contrast differ significantly across patients. We experimentally show that using this global meta-label by itself may improve segmentation performance, however it is more useful when combined with other meta-labels like slice partition. Our third meta-label for the ACDC dataset is the cardiac cycle phase (i.e., ES or ED). A single cycle of cardiac activity can be divided into two basic phases, the diastole where the ventricles are relaxed (not contracting), and the systole where the left and right ventricles contract and eject blood into the aorta and pulmonary artery, respectively. The first two rows of Fig. 2 show images corresponding to these two phases for the same patient. One can see that the size of the left and right ventricles changes considerably, and is much smaller at the end of the systole phase (ES). As shown in our experiments, incorporating information on the cardiac phase into the network's encoder generally helps improve segmentation quality.

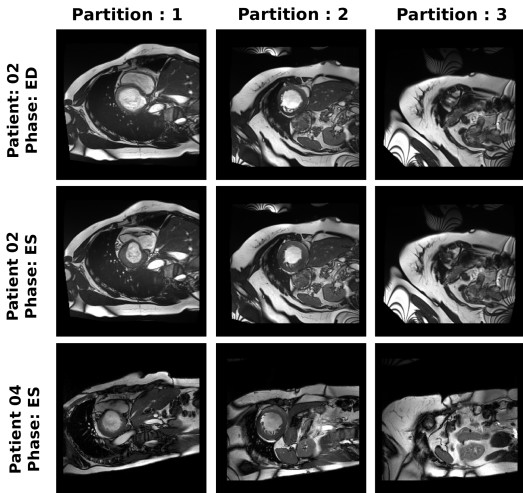

Figure 2: Examples of ACDC images with meta-labels. Volumetric images are sliced thought the short-axis and split into $S=3$ partitions. One can see that different slices in the same partition can share similar structure across volumes. We also consider the patient identity and the cardiac cycle phase as global meta-labels to guide the model optimization.

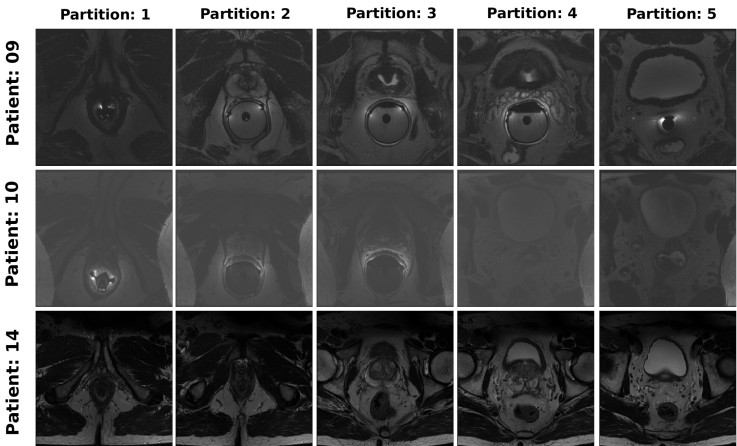

Figure 3: Examples of PROMISE12 images with their meta-labels. Slice partition and patient identity are used as the two meta-labels. We fixed the number of partitions to $S=5$. One can see a smooth transition on anatomical structures from a partition to the next. Note that this dataset exhibits high appearance variability across different patients.

## 3.1 Evaluation metric

We used the commonly-adopted Dice similarity coefficient (DSC) metric to evaluate segmentation quality of the tested methods. DSC measures the overlap between the predicted labels ($S$) and the corresponding ground truth labels ($G$):

$$\text{DSC}(S, G) = \frac{2 \, |S \cap G|}{|S| + |G|} \tag{1}$$

DSC values range from 0 to 1, a higher value corresponding to a better segmentation. In all experiments, we reconstruct the 3D segmentation for each scan by aggregating the predictions made on 2D slice and report their 3D DSC metric on the test set.

## 3.2 Experimental details

We assessed the performance of our self-paced contrastive learning approach when used in two different stages: **pre-training** and **semi-supervised learning**. The pre-training stage consists in optimizing

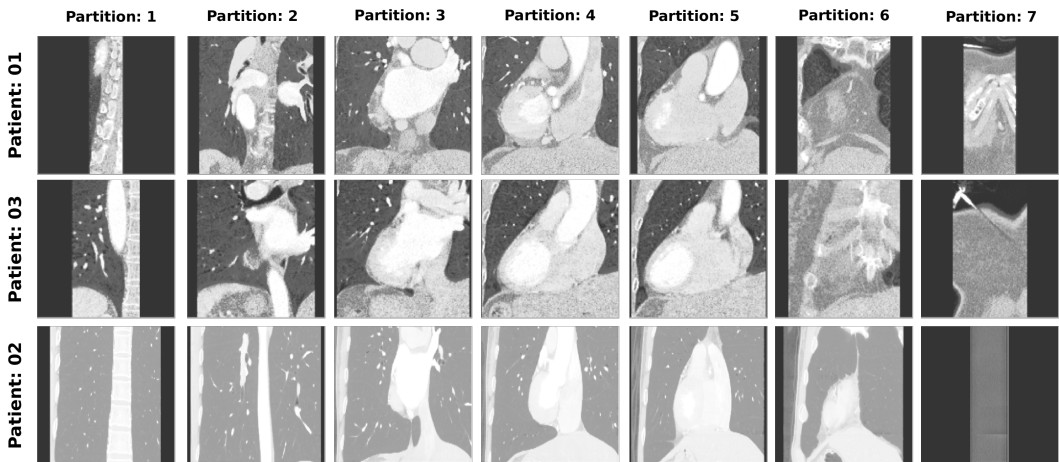

Figure 4: MMWHS images with their meta-labels. Slice partition and patient identity are also used as the two meta-labels. We fixed the number of partitions to $S = 7$. Smooth transition on anatomical structures can be observed from a partition to the next. This dataset also exhibits high appearance variability across different patients.

the encoder of a segmentation network on all available images via the contrastive loss. An additional fine-tune step is usually appended to this setting, which trains the whole network on a few labeled scans. In contrast, semi-supervised learning trains the network jointly with both labeled and unlabeled images. In both cases, we applied a learning rate warm-up strategy to increase the initial learning rate $lr_{\text{int}}$ by a factor of $N$ in the first 10 epochs, and then decrease it with a cosine scheduler for the following `max_epoch` $- 10$ epochs, as shown in Fig. 5.

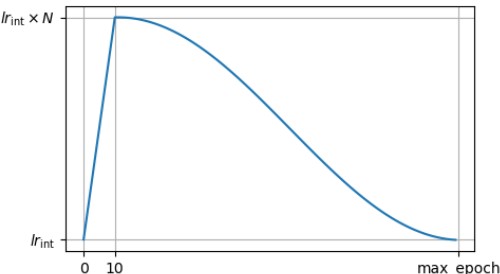

Figure 5: Learning rate warm-up and decay strategy used in our experiments

For pre-training stage, $lr_{\text{int}}$ was set to $5 \times 10^{-5}$, $N$ as 400 and `max_epoch` as 80 for the PROMISE12, MMWHS, Hippocampus and Spleen datasets, whereas we used `max_epoch` $= 120$ for ACDC. For the semi-supervised training stage or the fine-tuning procedure, we set $lr_{\text{int}}$ to $1 \times 10^{-7}$ for ACDC, Hippocampus, and Spleen datasets, $5 \times 10^{-7}$ for PROMISE12, and $2 \times 10^{-6}$ for MMWHS, with $N$ being set to 300 for all datasets. We fixed `max_epoch` to 75 for ACDC dataset and to 80 for the remaining datasets.

The generation of mini-batches is crucial for the contrastive learning. Following [3], we randomly sampled 10 scans from ACDC and drew one image per partition for each scan, resulting in 30 images per iteration. However, for the two other datasets, we randomly sampled 30 scans without considering meta-labels. We used a supervised loss $\mathcal{L}_{\text{sup}}$ to guide the network optimization during both the fine-tune procedure (for evaluating the quality of the pre-trained weights) and the semi-supervised training. Although different loss functions can be considered, such as the Dice loss [16] and Tversky loss [10], we adopted the well-known cross entropy loss in all experiments. This loss is defined as

$$\mathcal{L}_{\text{sup}} = -\frac{1}{|\mathcal{D}_l|\,|\Omega|} \sum_{(\mathbf{x},\mathbf{y}) \in \mathcal{D}_l} \sum_{i \in \Omega} \sum_{c=1}^{C} y_{ic} \log p_{ic}(\mathbf{x}) \qquad (2)$$

where $\mathcal{D}_l$ is labeled dataset, $\Omega$ is the 2-D pixel space and $p_{ic}(\mathbf{x})$ is the probability of class $c \in \{1, \dots, C\}$ predicted by the network at pixel $i$ .

For our contrastive pre-training experiments, we take the feature maps $E(\mathbf{x})$ at the end of the encoder and, following [5, 3], project them to low-dimensional vectors via a projector $g$ which consists of an average pooling to flatten the output of the encoder followed by a two-layer MLP to have a final dimensionality of 256.

Our proposed approach and compared methods have several tunable hyper-parameters, such as the weighting coefficient for each loss, the evolution strategy for $\gamma$ from Equ. (16) of the main paper, etc. Those hyper-parameters were chosen based on the performance of the method on the validation set. We ran our experiments on a computing cluster with Nvida P-100 GPUs. Running the pre-training stage usually takes less than 4 hours for all datasets, while the semi-supervised learning can take $4 - 6$ hours depending on the approach.

For our experiments shown in Tables 1, 2 and 3 of the main paper, we fixed the hyper-parameters as follows. The temperature $\tau$ was set to 0.07, close to related works [4, 13, 3] and working fine across different datasets. For $\lambda$ in Equ. (14), we used $\lambda_1 = 1.0$, $\lambda_2 = 1 \times 10^{-3}$ and $\lambda_3 = 1.0 \times 10^{-2}$ for the ACDC dataset, while values of $\lambda_1 = 1.0$, $\lambda_2 = 1 \times 10^{-1}$ were selected for the PROMISE12 and MMWHS dataset. For our combined model of Equ. (15), which involves a supervised loss, a regularization loss (Mean Teacher) and our SP-based contrastive loss, we fixed $\lambda_{\mathrm{reg}}$ to 0.1 and $\lambda_{\mathrm{sp}}$ to $5 \times 10^{-3}$ for the ACDC dataset, whereas $\lambda_{\mathrm{reg}}$ was set to 0.2 and $\lambda_{\mathrm{sp}}$ to $2 \times 10^{-2}$ for both the Prostate and MMWHS datasets. We also include a hyper-parameter sensitivity analysis of $\lambda$ in Sec. 4.6 to show the contribution of each loss component.

## 3.3  Details of compared methods

In Table 2 and 3, we compare our proposed self-paced contrastive method against several baselines, ablation variants of our method, and recently-proposed approaches for semi-supervised medical segmentation. We give a description of tested approaches below:

– **Contrastive loss [3]:** This setup evaluates the performance of a normal contrastive learning loss $\mathcal{L}_{\mathrm{con}}$ as in Equ. (2) of the main paper. In this case, we only consider the encoder features without our self-paced learning strategy to adapt the importance of positive pairs in the contrastive loss.

– **Self-Paced Contrastive Loss (SP-Con):** This is our full formulation of Equ. (3) in the main paper which exploits meta-labels for computing $\mathcal{L}_{\mathrm{SP}}^k$, where $k \in \{1, \dots, K\}$. As explained in Section 5.2, this loss can be used for pre-training as well as for semi-supervised learning. In our experiments, we test different combinations of this loss and other semi-supervised approaches.

– **Mean Teacher [11]:** This powerful method for semi-supervised learning adopts a teacher-student framework, where two networks sharing the same architecture learn from each other. Given an unlabeled image $\mathbf{x}$, the student model $p^s(\cdot)$ seeks to minimize the prediction difference with the teacher network $p^t(\cdot)$ whose weights are a temporal exponential moving average (EMA) of the student's:

$$\mathcal{L}_{\mathrm{MT}} = \frac{1}{|\mathcal{D}_u|\,|\Omega|} \sum_{\mathbf{x} \in \mathcal{D}_u} \sum_{i \in \Omega} \sum_{c=1}^{C} \left( p_{ic}^t(\mathbf{x}) - p_{ic}^s(\mathbf{x}) \right)^2 \tag{3}$$

Following the standard practice, we fix the decay coefficient to 0.999. The coefficient balancing the supervised and regularization losses is selected by grid search, from $1 \times 10^{-4}$ to 10.0.

– **Entropy Minimization (Entropy Min.) [12]:** This method, which has been successfully applied in semi-supervised classification [6] and segmentation [12] with domain gap, imposes a low conditional entropy on unlabeled images:

$$\mathcal{L}_{\mathrm{ent}} = -\frac{1}{|\mathcal{D}_u|\,|\Omega|} \sum_{\mathbf{x} \in \mathcal{D}_u} \sum_{i \in \Omega} \sum_{c=1}^{C} p_{ic}(\mathbf{x}) \log p_{ic}(\mathbf{x}). \tag{4}$$

By increasing its confidence for unlabeled images, the network pushes the decision boundary away from dense regions of the input space, thereby improving generalization. For this method, we performed a hyper-parameter search on the coefficient balancing the two losses, $\mathcal{L}_{\mathrm{sup}}$ and $\mathcal{L}_{\mathrm{ent}}$, from $1 \times 10^{-5}$ to $1 \times 10^{-1}$.

– **MixUp [14]:** We also evaluated the effectiveness of mixup, an effective data argumentation strategy on medical image segmentation, following [3].

– **Adversarial Training [15]:** This semi-supervised segmentation method trains a segmentation net-work and a classifier-based discriminator jointly in a min-max game. Its core idea is to enforce the segmentation predictions on unlabeled images being indistinguishable from those of labeled images, thus aligning the output distributions between labeled and unseen images. This method works particularly well in a scenario where scans present large variability causing a domain gap.

– **Discrete Mutual Information maximization [8]:** This semi-supervised segmentation technique maximizes the mutual information between two sets of feature maps undergoing different geo-metric transformations. These feature maps are taken from different hierarchical levels of the segmentation network, thus regularizing the network at multiple scales. It was shown effective for segmenting medical images with limited supervision. In this experiment, we optimize the mutual information between features taken from both the encoder (as our proposed method), and from the decoder.

– **Global and Local Contrastive [3]:** This last approach is our full implementation of [3], which takes into account not only the encoder's features as a global descriptor, using Equ. (2) of the main paper, but also dense features from decoder blocks that allow an effective contrastive learning at the pixel level. For the decoder, we chose the output of the third decoder block and resized the dense features to a fixed resolution of $10 \times 10$ pixels with adaptive average pooling. For each feature map, we then randomly sampled 5 different spatial locations and performed contrastive learning using Equ. (1) of the main paper on these $5 \times 2N_{\text{batch}}$ vectors, where $N_{\text{batch}}$ is the number of images in the current batch. Following [3], we adopted a two-step pre-training strategy, and pre-trained the decoder while freezing the encoder. Using the well pre-trained weights, we report the test score when fine-tuning models on a few labeled scans.

## 4 Additional experimental results

### 4.1 Supplementary experiments on two extra datasets

The proposed method could work with any segmentation task where meta-labels are available. This includes segmenting volumetric data of any organ for which a rough correspondence can be obtained between 2D slices in the volume. In order to further highlight the robustness of our proposed method, we carried out additional experiments on two extra datasets segmenting the hippocampus and spleen from MRI and CT images. For these two dataset, we only tested our proposed variant SP-Con (*pre-train*) on the slice position meta-label ($\mathcal{L}_{\text{sp}}^1$) and compared it against strong concurrent approaches such as Contrastive ($\mathcal{L}_{\text{con}}^1$) [3] and Mean Teacher [11].

As one can see from Table 1, our proposed method outperforms Contrastive in most cases and reaches a performance comparable with Mean Teacher. This confirms the general usefulness of our self-paced learning strategy for the semi-supervised segmentation of different organs.

Table 1: 3D DSC performance of the proposed SP-Con (*pre-train*) variant using slice position as meta-label and other approaches on hippocampus and spleen datasets with few labeled scans.

| Method | Pretrain | Train | Hippocampus | | | Spleen | |
|---|---|---|---|---|---|---|---|
| | | | 1 scan | 2 scans | 4 scans | 2 scan | 4 scans |
| Baseline | - | $\mathcal{L}_{\text{sup}}$ | 60.87 | 73.33 | 78.82 | 65.51 | 68.59 |
| Mean Teacher [9] | - | $\mathcal{L}_{\text{sup}} + \mathcal{L}_{\text{mt}}$ | **70.06** | 75.65 | 80.60 | 55.02 | 68.10 |
| Contrastive [3] | $\mathcal{L}_{\text{con}}^1$ | $\mathcal{L}_{\text{sup}}$ | 64.40 | 75.00 | **81.45** | 65.14 | 67.21 |
| SP-Con (*Pre-train*) | $\mathcal{L}_{\text{sp}}^1$ | $\mathcal{L}_{\text{sup}}$ | 66.70 | **76.89** | 81.25 | **69.05** | **69.64** |

### 4.2 Fine-tuning the model on the whole dataset with annotations

We showed previously that pre-training a network with images and meta-labels from the dataset helps to improve performance when fine-tuning it using a few labeled examples. In this section, we consider the following question: "is our pre-training loss helpful when fine-tuning the model on the entire dataset with ground-truth annotations, instead of just a few of them?" To answer this question, we first pre-trained a network with our SP-Con (*pre-train*) loss and fine-tuned it with different numbers of labeled images on the ACDC dataset. Table 2 summarizes the segmentation performance of our method after fine-tuning, and compares it to Mean Teacher.

Table 2: 3D DSC improvements brought by SP-Con (Pre-train) with slide position as meta-label with various numbers of labeled scans.

| Methods | 1 scan | 2 scans | 4 scans | 8 scans | 175 scans |
|---|---|---|---|---|---|
| Baseline | 57.53 | 67.06 | 75.64 | 82.64 | 88.06 |
| Mean Teacher | 62.85 | 72.84 | 79.12 | 84.35 | N/A[2] |
| SP-Con (*pre-train*) | 73.99 | 81.01 | 82.83 | 84.29 | 88.35 |
| Our Improvement | 16.46 | 13.95 | 7.18 | 1.65 | 0.29 |

We observe that our method's relative improvement with respect to the baseline reduces as the number of labeled samples increases. This is expected since there is no additional unlabeled data to exploit when using the entire set of images as labeled data (175 scans). However, SP-Con still yields a small improvement (0.29%) compared to the full supervision (Baseline with 175 scans as labeled images).

### 4.3 Segmentation performance when randomly selecting labeled volumes per experiment

For the previous experiments, when fine-tuning the model or using a semi-supervised training with a few labeled data, we kept a fixed split on the labeled/unlabeled images for each number of labeled scan, and reported results were obtained by averaging performance from three independent runs with different random seeds controlling model initialization, data augmentation randomness, and data fetch ordering. In order to remove the possible bias from the choice of these fixed splits, we ran three additional experiments using different labeled/unlabeled splits per experiment, the results of which are reported in Table 3.

Table 3: 3D DSC performances for different splits on the ACDC dataset. Best cases are highlighted in **bold**.

| Method | Pretrain | Train | Split 1 | | Split 2 | | Split 3 | | Mean (std) over splits | |
|---|---|---|---|---|---|---|---|---|---|---|
| | | | 1 scan | 2 scans | 1 scan | 2 scans | 1 scan | 2 scans | 1 scan | 2 scans |
| Baseline | - | $\mathcal{L}_{sup}$ | 61.36 | 71.11 | 35.76 | 71.48 | 39.74 | 76.44 | 45.62 (11.25) | 73.01 (2.43) |
| Unsup. Con. | $\mathcal{L}_{con}^{unsup.}$ | $\mathcal{L}_{sup}$ | 65.70 | 77.84 | 50.69 | 72.87 | 59.65 | 80.22 | 58.68 (6.17) | 76.98 (3.06) |
| Unsup. Con. + SP | $\mathcal{L}_{sp}^{unsup.}$ | $\mathcal{L}_{sup}$ | 69.34 | 77.49 | 61.96 | 73.11 | 66.55 | 80.82 | 65.95 (3.04) | 77.14 (3.16) |
| Contrastive | $\mathcal{L}_{con}^1$ | $\mathcal{L}_{sup}$ | 72.56 | **80.58** | 67.68 | 73.97 | 68.64 | 82.35 | 69.63 (2.11) | 78.97 (3.61) |
| | $\mathcal{L}_{con}^2$ | | 68.19 | 76.69 | 43.69 | 71.57 | 57.34 | 80.31 | 56.41 (10.02) | 76.19 (3.59) |
| | $\mathcal{L}_{con}^3$ | | 65.40 | 76.33 | 64.93 | 69.10 | 58.82 | 80.06 | 63.05 (3.00) | 75.16 (4.55) |
| Mean Teacher | - | $\mathcal{L}_{mt}$ | 73.04 | 78.97 | 60.91 | 72.82 | 55.07 | 80.26 | 63.01 (7.48) | 77.35 (3.25) |
| SP-Con (*pre-train*) | $\mathcal{L}_{sp}^1$ | $\mathcal{L}_{sup}$ | **76.24** | 79.96 | **68.18** | **76.46** | **74.18** | **82.46** | **72.87 (3.42)** | **79.63 (2.46)** |
| | $\mathcal{L}_{sp}^2$ | | 71.77 | 80.08 | 58.07 | 71.95 | 58.02 | 81.07 | 62.62 (6.47) | 77.70 (4.09) |
| | $\mathcal{L}_{sp}^3$ | | 66.77 | 77.10 | 63.65 | 73.41 | 62.03 | 82.11 | 64.15 (1.97) | 77.54 (3.57) |

In this new set of experiments, we followed the same experimental protocols as in previous one, where we first pre-train the network with different contrastive losses (unsupervised, contrastive with three meta-labels, and those with self-paced learning) and then fine-tune them using a few labeled scans. However, this time vary the scans used as labeled data for each run and compute the final performance by averaging results from the three experiments.

As one can see from Table 3, using different splits for the labeled data leads to a large variance in performance. When given a single labeled scan, the baseline DSC for different splits varies from 35.75% to 61.36%, with a standard derivation up to 11.25%. On the other hand, our model using meta-labels obtains more robust results (e.g., standard deviation of 3.42% for $\mathcal{L}_{sup}^1$ using one labeled scan) and outperforms the contrastive counterparts in all but one cases.

### 4.4 Self-paced learning analysis

As discussed in Section 5.1 of the main paper, we include the self-paced weights $w_{ij}$ taken from different epochs, corresponding to four different $\gamma$ (marked as orange star) in Fig. 1 of the main paper. Notice that $w_{ij}$ is only defined on positive pairs, however we visualize $w_{ij}$ for all image

---

[2]Mean Teacher requires unlabeled examples for its consistency loss, thus was not considered for the full supervision setting.

pairs to verify whether our proposed self-paced strategy can successfully learn weights that capture the corresponding meta-label's quality. We train our proposed loss using slice position as the only meta-label (i.e., $\mathcal{L}_{sp}^1$).

Figure 6: Self-paced importance weight $w_{ij}$ for two scans during the optimization. We plot not only the $w_{ij}$ for positive pairs, but also for negative pairs.

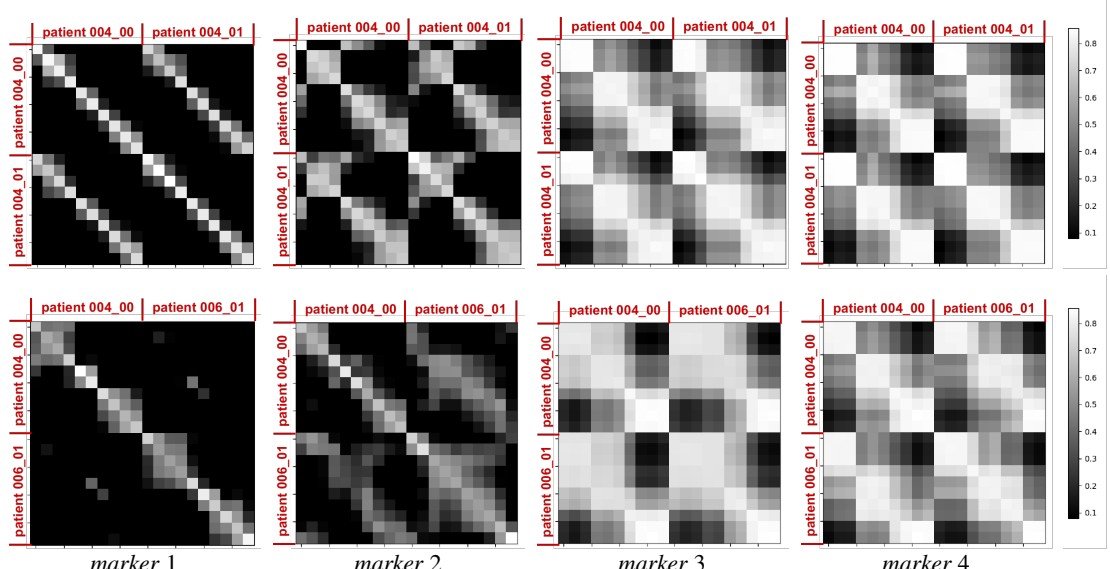

In Fig. 6, we first compare two scans from the same patient with different cardiac phases (first row of the figure). As can be seen, in the beginning of the training (*marker* 1), self-paced weights have a high value for pairs of slices from the same position, even when corresponding to different phases of the cardiac cycle. As the training progresses, our proposed strategy learns to assign relatively high values on positive pairs, and keep low values for negative pairs (*marker* 2). At the end of the training, sharper blocks form in weight matrix based on meta-label classes, and the network learns to distinguish different slice partitions (*marker* 3-4). In contrast, conventional contrastive loss lacks the ability to learn gradually and does not have an effective selection mechanism in the beginning of training. We also conducted a similar analysis with two scans taken from different patients (second row of the figure). It can be seen that the proposed loss learns to assign weights based on the network's uncertainty for each pair. We can observe that the network is slower to assign high values to positive pairs across patients in the first few epochs, since they differ in terms of appearance.

### 4.5 Sensitivity to $\gamma$ parameters

The next experiment investigates the impact of $\gamma_{start}$ and $\gamma_{end}$ in Equ. (16) of the main paper, where we fixed $p = 1/2$ and adopt the linear strategy to ramp-up $\gamma$. Theorem 2 of the main paper states that $\ell_{ij}$ is bounded by $\log 2N + 2/\tau$. However, for the linear strategy, where $w_{ij} = 1 - \frac{1}{\gamma}\ell_{ij}$, we found that further increasing $\gamma$ boosts the segmentation performance. Table 4 shows the results of a grid search varying these two hyper-parameters ($\gamma_{start}$ in rows and $\gamma_{end}$ in columns) and measuring the performance on the validation set. As can be seen, the segmentation DSC reaches its maximum value when $\gamma_{start}$ is 2.0. Given a fixed ramp-up strategy, choosing a small $\gamma_{start}$ causes the learning to ignore most of the examples in the beginning of the training and, thus, results in under-fitting. In contrast, a large value of $\gamma_{start}$ pushes the network to treat all examples equally. Since $\gamma_{end}$ controls the level of weights in the end of the training, it has a similar behavior as $\gamma_{start}$.

### 4.6 Sensitivity to $\lambda_k$ parameters

We then present a sensitivity analysis for $\lambda_k$ in Equ. (14) on the ACDC dataset. As $\mathcal{L}_{sp}^1$ (the one using slice partition as the supervision signal) resulted in the best performance across all dataset, we fixed $\lambda_1$ as 1.0 and performed a grid search on $\lambda_2$ and $\lambda_3$ with logarithmic scales ranging from 0.1 to

Table 4: Sensitivity analysis of hyper-parameter $\gamma$.

| $\gamma_{\text{start}}$ | $\gamma_{\text{end}}$ | | | | |
|---|---|---|---|---|---|
| | 40.0 | 50.0 | 60.0 | 70.0 | 80.0 |
| 1.0 | 72.65 | 73.83 | 72.98 | 72.60 | 72.99 |
| 1.5 | 72.56 | 73.95 | 74.97 | 72.56 | 73.79 |
| 2.0 | 73.20 | **75.08** | 73.32 | 73.39 | 72.70 |
| 2.5 | 72.61 | 73.75 | 72.26 | 72.49 | 73.20 |
| 3.0 | 72.95 | 73.50 | 72.89 | 73.96 | 73.33 |
| 3.5 | 73.62 | 73.08 | 74.28 | 74.38 | 73.06 |

0.001. We report the 3D DSC performance on the validation set using just one scan as labeled data, as shown in Table 5. We see that varying both $\lambda_2$ and $\lambda_3$ leads to relatively stable performance for the downstream segmentation tasks, while the best performance is achieved by setting $\lambda_2 = 0.01$ and $\lambda_3 = 0.001$. This suggests that, for the ACDC dataset, using Patient Identity as the meta-label brings a more useful information compared to Cardiac cycle phase.

Table 5: Sensitivity analysis of hyper-parameters $\lambda_2$ and $\lambda_3$ for the ACDC dataset.

| $\lambda_2$ | $\lambda_3$ | | |
|---|---|---|---|
| | 0.1 | 0.01 | 0.001 |
| 0.1 | 73.09 | 74.12 | 74.61 |
| 0.01 | 73.99 | 74.56 | **74.89** |
| 0.001 | 73.52 | 74.38 | 74.52 |

## 4.7 Visual result inspection

Last, we provide a visual inspection on segmentation predictions for different approaches on the three first datasets. As can be seen, the proposed method effectively enhances the segmentation quality on the different datasets, and reaches a segmentation prediction closer to the ground truth than other tested approaches.

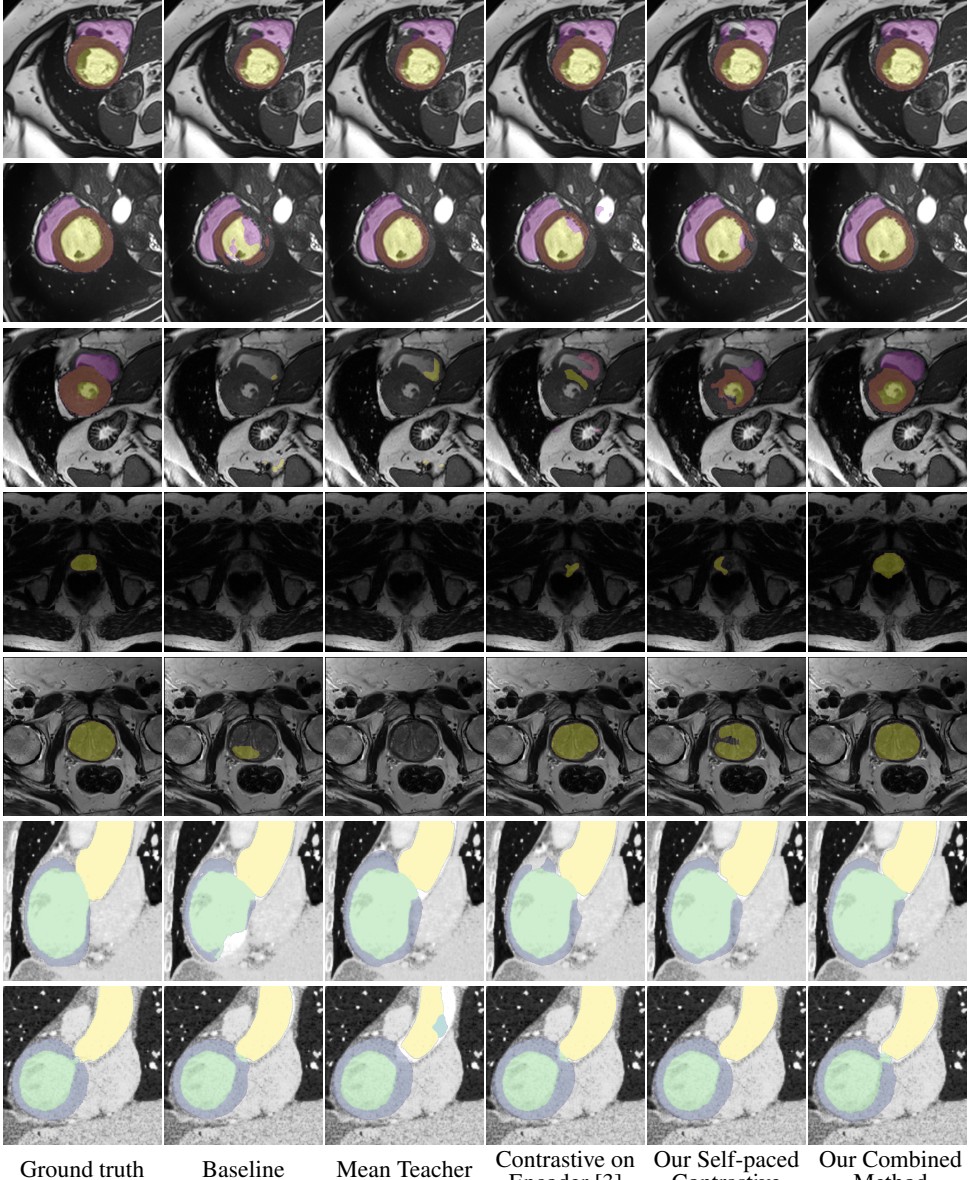

|  |  |  | Contrastive on | Our Self-paced | Our Combined |
| Ground truth | Baseline | Mean Teacher | Encoder [3] | Contrastive | Method |

Figure 7: Visual comparison of tested methods on test images. **Rows 1–3**: the ACDC; **Rows 4–5**: the PROMISE12; **Rows 6–7**: MMWHS.