# OpenReview forum: "Self-Paced Contrastive Learning for Semi-supervised Medical Image Segmentation with Meta-labels"
_NeurIPS.cc/2021/Conference — NeurIPS 2021 Poster_

### Official Review · Reviewer_UfTw · 2021-07-14

**Rating:** 7
**Confidence:** 3

**Summary:**

In this work, the authors show how to use weak labels to boost segmentation of medical datasets.
The idea hinges on a combination of contrastive learning with a new self-paced learning approach that gives preference to more confident labels (and therefore not being mislead by label noise or irrelevant weak labels).

**Ethical Concerns:**

Nothing that goes beyond the concerns one could voice about virtually all deep learning submissions these days.

**Limitations And Societal Impact:**

A social impact and limitations sections exists and I agree with the authors comments in it.

**Main Review:**

I enjoyed reading the paper. It is written very clearly, addresses very relevant issues (learning with limited data, utilisation of weak labels), and convinced me by presenting a solid set of experiments as well as ablations/baselines.

The theoretical contributions are, as they are presented, useful and novel. Contributions might not be seen as earth shattering, are likely not the start of a new paradigm, but can still be labeled as solid. The main text points at all hyper-parameters and how good values have been found (grid search, mostly). The range for sensible $\gamma$ values, one of the more critical parameters to choose, are discussed in detail, and presented understandably.

While I'm fairly confident that I understand the manuscript very well, I want to point out that I do not know the related work very well and would not know if some work exists that was not mentioned/cited by the authors.

In summary, the manuscript presents a solid method that shows excellent performance wrt. to competing methods, even if only very limited training data is available. Contributions are good, presentation excellent.

Minor comments:
line 144 -- I would have preferred to find a more complete list of used meta-labels.
line 159 -- What is g(.)? Could the authors be more concrete about it?
line 170 -- Superfluous "A".
line 182 -- missing "y" in "by".
page 5 -- l_{i,j} is used with inconsistent font
line 198 -- shouldn't it be l_{i,j} <= \gamma ???
line 218 -- I was wondering how critical is it at the end to find the right \lambda_k? (How fine do you need to grid search, how far can you be off before results drop significantly in quality?)


**Time Spent Reviewing:**

about 5 hours in 3 sessions distributed over one day

---

> ### Author Response · Authors · 2021-08-11
> **Answer to Reviewer UfTw**
>
> Thank you for your time and insightful feedback!
>
>
> We will incorporate the changes into the revised paper, and address the issues in detail here:
>
>
> _Line 104, more complete list of used meta-labels_:
>
> We present all the used meta-labels in the section of results (l.304). However, we agree that listing them even before can help to better understand the paper, and will modify the paper accordingly
>
>
> _Line 159 -- What is g(.)_:
>
> For the non-linear projection g(.) we used the same methodology as in previous contrastive learning papers (e.g.[1]). It consists of an average pooling to flatten the output of the encoder E(x) and adding a two layers perceptron to lead to a final dimensionality of 256. We will add more details in the supplementary material.
>
> [1]: Chen, Ting, et al. "A simple framework for contrastive learning of visual representations." International conference on machine learning. PMLR, 2020.
>
>
> _line 198 -- shouldn't it be_ $l_{i,j} <= \gamma$:
>
> Yes, thanks, you are right. We will correct this error in the reviewed version of the paper.
>
> _how critical is the right_ $\lambda_k$:
>
> Here we present a sensitivity analysis for $\lambda$ on the ACDC dataset. As $L_{\mathrm{SP-con}}^{1}$ (the one using slice partition as the supervision signal) results in the best performance, we fixed $\lambda_1$ as 1.0 and performed a grid search on $\lambda_2$ and $\lambda_3$ with logarithmic scales ranging from 0.1 to 0.001. We report the 3D Dice performance on the validation set the ACDC dataset using just 1 scan as labeled data:
>
>
> |                  | $\lambda_2=0.1$ | $\lambda_2=0.01$ | $\lambda_2=0.001$ |
> |------------------|----------------|-----------------|------------------|
> | $\lambda_3=0.1$   | 73.09          | 74.12           | 74.61            |
> | $\lambda_3=0.01$  | 73.99          | 74.56           | 74.89            |
> | $\lambda_3=0.001$ | 73.52          | 74.38           | 74.52            |
>
>
> From the table, we see that when $\lambda_2$ and $\lambda_3$ have small values, the performance is relatively stable.

---

> > ### Comment · Reviewer_UfTw · 2021-08-17
> > **Thanks**
> >
> > Thanks for your answers. I see no reason to change my evaluations and I still think this paper is fit for publication at NeurIPS.

---

### Official Review · Reviewer_n24V · 2021-07-15

**Rating:** 6
**Confidence:** 2

**Summary:**

The authors propose using “metal labels” in pertaining image encoders on unlabeled datasets. They use the metal-labels in pertaining the encoder, and also as an auxiliary task in training. They show their method works on multiple medical datasets.

**Ethical Concerns:**

No concerns

**Limitations And Societal Impact:**

Limitations are short - I’ve mentioned a number above, but these are not discussed in the limitations section.

The social impact of the problems tackled is clear, as the authors have focused on medical use cases. I don’t think it is as hard to collect unlabeled medical data as they are claiming (there are many large open source datasets), but the quality of labels is poor so the problem remains an important one.

**Main Review:**

The value of clustering 2D slices in a 3D volume seems not as helpful as just learning the model on 3D data in the first place (a 3D implementation of the U-net exists).

More information could be provided on what meta-labels were included. It’s possible that manually viewing them with some domain expertise could already select those that are unlikely to contribute any signal.

Best performance is only achieved by combining everything, compared with the baselines that quite possibly focus on a single innovation. Compared with the other individual methods tried, the best performance is achieved by the baselines in some cases. So the paper becomes more about how much one can push performance than introducing a single innovation.

**Time Spent Reviewing:**

1

---

> ### Author Response · Authors · 2021-08-11
> **Answer to Reviewer n24V**
>
> Thank you for your time and insightful feedback!
>
> We will incorporate the changes into the revised paper, and address the issues in detail here:
>
>
> _2D slices vs. 3D volume_:
>
>  2D-based networks are widely used in medical image segmentation. [1] reports results for medical image segmentation on several datasets and show that 2D segmentation is on par or better than their 3D model. This can be partially explained by the fact that, for certain datasets (e.g. ACDC dataset), x-y images can be misaligned (e.g., due to cardiac movement in ACDC) and therefore performing a 3D segmentation can lead to lower segmentation accuracy. Additionally, 2D-based networks also have a lower computational cost and faster inference speed.
>
> [1] Baumgartner, Christian F., et al. "An exploration of 2D and 3D deep learning techniques for cardiac MR image segmentation." International Workshop on Statistical Atlases and Computational Models of the Heart. Springer, Cham, 2017.
>
>
> _More information about meta-labels_:
>
> Thanks for the comment. In the paragraph starting from line 304, we report which meta-labels are used in the experiments. More information is also provided in the Datasets section in Supplementary material. Moreover, in Section 2 (and Fig.1-3) of the supplementary material, we compare images with the same or different meta-labels.
>
>
> _Is it possible to select the useful meta-labels manually by inspection_:
>
> We did not try any manual selection of the meta-labels. From the results, we see that in general, slice location is the most informative meta-label that helps the most to improve results. However, the best results are obtained when combining all meta-labels available for a given dataset (see table 2).
>
>
> _Best performance is only achieved by combining everything_:
>
> In Table 2, “SP-con(both) L1” is our simplest implementation of the proposed approach where we use pre-training and semi-supervised learning with the contrastive loss based on a single meta-label together with self-paced learning (which are our main contributions). This configuration is already on par with or better than most of the other approaches in table 3. Hence, we argue that our paper offers both 1) a general technique based on self-paced contrastive learning that can boost performance in various segmentation tasks, and 2) state-of-the-art performance when the proposed technique is combined with powerful methods for semi-supervised learning like Mean Teacher.
>
>
> _Limitations are short_:
>
> We answered all reviewer questions above. We will include those either in the main paper (difficult due to page limits) or in the supplementary material.
>
>
> _I don’t think it is as hard to collect unlabeled medical data_:
>
> As medical data might have very different modalities, we still believe that it is difficult to have large datasets of unlabelled data for all modalities. Additionally, it is difficult to share medical data because of privacy issues.

---

> > ### Comment · Reviewer_n24V · 2021-08-23
> > **Thanks**
> >
> > I thank the authors for their considered response. I would like to upgrade my rating to "6: Marginally above the acceptance threshold" on the basis of their response.

---

### Official Review · Reviewer_G3GS · 2021-07-19

**Rating:** 6
**Confidence:** 4

**Summary:**

This work proposes the use of meta labels and self paced learning to improve performance of medical image segmentation when using contrastive pre-training. Through results on three different medical image segmentation tasks, the authors demonstrate the efficacy of the proposed method.


**Limitations And Societal Impact:**

Major concern regarding the work is
- why the experimentation setup doesn't exactly match https://arxiv.org/pdf/2006.10511.pdf? The papers seem to use different number of training examples
- it would be good to see the demonstration of self paced learning beyond segmentation or medical tasks. Without the results, it is difficult to ascertain the generality of the proposed method(s)

If the authors can address these concerns satisfactorily, will be happy to reconsider my scores.

**Main Review:**

Originality:
While the paper is mostly built on top of https://arxiv.org/pdf/2006.10511.pdf, the addition of self paced learning and meta labels are interesting improvements on top of the prior work.

Quality and clarity:
The paper is mostly well-written and easy to follow. The authors demonstrate good experimental rigor with experiments on multiple datasets and good ablations

Significance:
Its challenging and expensive to obtain labels for medical segmentation. Efforts like this which aim to improve the sample efficiency for medical segmentation are great research directions in general

Detailed Comments:

> However, thanks to large datasets like ImageNet [11], the amount of labeled data for training has increased and, in such setting, pre-training often hinders performance [36].

This comment is a bit confusing. You may want to clarify that while supervised pre-training on Imagenet is still the norm for developing many vision models, this comment is for unsupervised pre-training

> Contrastive training as a regularization loss in semi-supervised learning.

Please cite related work - https://arxiv.org/abs/2102.06605
There is a lot of recent work on self supervised learning for medical imaging and segmentation. Related work section is missing those papers.
Eq 2 is pretty much an instantiation of the supervised contrastive objective. Please cite this in the text

>  These update rules in Equ. (7) can be explained intuitively. For a given γ, the hard threshold rule only considers confident pairs with Lij ≥ γ and ignores the others

There seems to be an error here and i think this should be Lij < y

>  contrastive learning has mostly been used for pre-training the model. Here, we show that it can further boost results in a semi-supervised setting.

Please cite https://arxiv.org/abs/2006.10029 here

> How is gamma in eq 16 related to the gamma derived from equation 2? It is not clear they relate.

Please substantiate the relationship here.

It is difficult to read the results in table 3. Please bold the best ones

> Please add more details as to how the self paced objective was in the semi-supervised setup without the contrastive objective.

Assume the loss in this case would be wrt individual training examples?


**Time Spent Reviewing:**

4

---

> ### Author Response · Authors · 2021-08-11
> **Answer to Reviewer G3GS**
>
> Thank you for your time and insightful feedback!
>
> We will incorporate the changes into the revised paper, and address the issues in detail here:
>
>
> _Comment about ImageNet pre-training_:
>
> Thanks for the comment. Indeed, pretraining on a large-scale labelled dataset usually results in a good performance on downstream tasks with small and moderate datasets. Recently, researchers have found that this paradigm may not be as powerful as expected when the downstream task has large annotated data [1, 2]. We will rephrase our statement to make it clearer.
>
> [1]: Zoph, Barret, et al. "Rethinking pre-training and self-training." arXiv preprint arXiv:2006.06882 (2020).
>
> [2]: He, Kaiming, Ross Girshick, and Piotr Dollár. "Rethinking imagenet pre-training." Proceedings of the IEEE/CVF International Conference on Computer Vision. 2019.
>
>
> _Contrastive training as a regularization loss in semi-supervised learning_:
>
> We will add [3] in our Related work section. This work uses the naive contrastive loss as a secondary loss complementary to cross-entropy in a fully supervised setting, thus is different from ours. We also found two new related works [4,5] on contrastive learning for medical image segmentation, which can also be added as references to our paper. We note that [4] and [5] were added to Arxiv after the Neurips submission deadline in May 2021.
>
> [3] Zhang, Yifan, et al. "Unleashing the Power of Contrastive Self-Supervised Visual Models via Contrast-Regularized Fine-Tuning." arXiv preprint arXiv:2102.06605 (2021).
>
> [4] Zeng, Dewen, et al. "Positional Contrastive Learning for Volumetric Medical Image Segmentation." arXiv preprint arXiv:2106.09157 (2021).
>
> [5] Pandey, Prashant, et al. "Contrastive Semi-Supervised Learning for 2D Medical Image Segmentation." arXiv preprint arXiv:2106.06801 (2021).
>
>
> _The sentence in line 198_:
>
> Thank you for finding the error in line 198. $\ell_{ij}$ should be smaller or equal than $\gamma$. We will correct the sentence.
>
>
> _Results in a semi-supervised setting_:
>
> Thanks for pointing out an important paper that we have missed. We will cite [6] in our revised version. In [6], the training is performed in two separate steps. First, a large teacher model is firstly trained with contrastive loss and then fine-tuned with a small fraction of labelled data. Second, a task-specific student model is trained based on the teacher's knowledge using supervised and conventional distillation losses. However, there is no joint optimization between supervised and contrastive objectives. Our method can achieve significant improvement by jointly optimizing them.
>
> [6] Chen, Ting, et al. "Big self-supervised models are strong semi-supervised learners." arXiv preprint arXiv:2006.10029 (2020).
>
>
> _How is gamma in eq 16 related to the gamma derived from equation 2_:
>
> We could not find any $\gamma$ in Equ.(2). Gamma appears only in Equ.(3) as a parameter of the regularization term $R_{\gamma}$. Equ.(16) defines how the $\gamma$ in Equ.(3) is updated during optimization.
>
>
> _It is difficult to read the results in table 3_:
>
> The best results in Table 3 are highlighted in bold colored font. We believe the reviewer is referring to Table 2, where the best results are not highlighted. If so, we will update Table 2 using the same highlighting scheme as Table 3.
>
>
> _Add more details of self-paced objective in the semi-supervised setup without the contrastive objective_:
>
> Thanks for the comment. The proposed self-paced learning strategy is used only in the contrastive loss. As specified in the supplementary material, in the semi-supervised setup, we employed the Mean Teacher and Entropy minimization methods which use Equ (15) without $L_{sp-con}$. Mean Teacher adopts a teacher-student framework and $L_{reg}$ is the L2 loss between teacher and student, while entropy minimization enforces uncertainty minimization by reducing the entropy of conditional predictions, and in this case, $L_{reg}$ is the entropy term.
>
>
> _Why the experimentation setup doesn't exactly match [7]_:
>
> Results in [7] are computed averaging over different random splits. As also shown in the following table for ACDC, that experimental protocol introduces a high variance in the results. Thus, we prefered to keep a single random split but varying the other random parameters such as random initialization, selected samples in SGD and random transformations. Still, we compare [7] with our setting in table 3.
>
>
> |                        | split1 |         | split2 |         | split3 |         | mean   |         |
> |------------------------|:------:|---------|--------|---------|:------:|---------|--------|---------|
> |                        | 1 scan | 2 scans | 1 scan | 2 scans | 1 scan | 2 scans | 1 scan | 2 scans |
> | Baseline               |  61.36 | 71.11   |  35.76 | 71.48   |  39.74 | 76.44   |  45.62 | 73.01   |
> | Unsup_Con              |  65.70 | 77.84   |  50.69 | 72.87   |  59.65 | 80.22   |  58.68 | 76.98   |
> | Unsup_Con + SP         |  69.34 | 77.49   |  61.96 | 73.11   |  66.55 | 80.82   |  65.95 | 77.14   |
> | Contrast + L1          |  72.56 | 80.58   |  67.68 | 73.97   |  68.64 | 82.35   |  69.63 | 78.97   |
> | Contrast + L2          |  68.19 | 76.69   |  43.69 | 71.57   |  57.34 | 80.31   |  56.41 | 76.19   |
> | Contrast + L3          |  65.40 | 76.33   |  64.93 | 69.10   |  58.82 | 80.06   |  63.05 | 75.16   |
> | Mean Teacher           |  73.04 | 78.97   |  60.91 | 72.82   |  55.07 | 80.26   |  63.01 | 77.35   |
> | SP-con (pretrain) + L1 |  76.24 | 79.96   |  68.18 | 76.46   |  74.18 | 82.46   |  72.87 | 79.63   |
> | SP-con (pretrain) + L2 |  71.77 | 80.08   |  58.07 | 71.95   |  58.02 | 81.07   |  62.62 | 77.70   |
> | SP-con (pretrain) + L3 |  66.77 | 77.10   |  63.65 | 73.41   |  62.03 | 82.11   |  64.15 | 77.54   |
>
>
> [7] Chaitanya, Krishna, et al. "Contrastive learning of global and local features for medical image segmentation with limited annotations." arXiv preprint arXiv:2006.10511 (2020).
>
>
> _Self paced learning beyond segmentation or medical tasks_:
>
> [8] shows with an extensive set of experiments on image classification that curriculum learning (and self-paced learning as a special case of it) works when data is noisy or when training on a budget. In our case, the used meta-labels are noisy and therefore using self-paced learning can help improve performance. Moreover, we believe the proposed method could also be useful for other computer vision tasks, such as classifying and segmenting non-medical images. We plan to explore this in future work.
>
> [8] Wu, Xiaoxia, Ethan Dyer, and Behnam Neyshabur. "When Do Curricula Work?." International Conference on Learning Representations (2021).

---

> > ### Comment · Reviewer_G3GS · 2021-08-23
> > **Thanks**
> >
> > I very much appreciate the authors thoughtful response to my review. Having carefully considered it, I prefer to keep my score.

---

### Official Review · Reviewer_HQGh · 2021-07-21

**Rating:** 6
**Confidence:** 4

**Summary:**

Annotation in the clinical dataset is hard to prepare, so it is important if self-learning algorithms can be applied to medical data which can do few-shot learning. In this paper, authors put forward a method to do segmentation which makes use of pretraining with contrastive learning with metadata, and then better fine tuning with self-paced learning. The second step is necessary to mitigate meta data noise. This paper shows experiments on three public datasets. With a few annotated data, the algorithms still show comparable performance with the fully supervised segmentation algorithm.

**Ethics Review Area:**

["I don’t know"]

**Limitations And Societal Impact:**

One weakness is in the experiment design. In this paper, table 2, authors showed the ablation study of all combination of loss functions on all three datasets, and selected “SP-Con (both) + Mean Teacher” as the best model to report. It would be better if this ablation study and model selection was done only with one of the three datasets, and only report the best model’s performance on the rest, so that it can follow the best practice of model selection. In table 1 (section 5.1) the selection of hard vs. linear self-paced regularization is a good example of parameter selection.
Another limitation for the current submission is that authors did not discuss whether all organs can be segmented with the proposed algorithm design. Right now, from the dataset point of view, it works with prostate (PROMISE12) and heart (ACDC and MMWHS). Further experiments and discussion with different organs could better support the idea that proposed method could have a good clinical impact.

Overall, this paper is well written. However, it could be better if an algorithmic workflow figure is provided. Right now, although this paper put much effort to discuss about the methodology, it is still not straight forward enough for readers to understand how this algorithm works from input to contrastive learning, and to the self-paced learning.
In table 1, performance on ACDC is shown with percentage. It would be better, if in the table title, the authors could explain what are these percentage numbers: are they dice or auc or any other metrics.

More graphics in the main body, instead of putting them in the supplementary material, could better help readers understand the idea. For example, choosing one row from Figure 1, Figure 2, and Figure 3 in the supplementary material respectively as a new figure, and moving it to the main body as examples of “meta-labels”. A selection of Figure 6 in the supplementary material could also be shown in the main body, to visually show the segmentation performance.

Additionally,
- There are many hyper-parameters used in the paper, which are determined using grid search on the validation set. But they are not specified, including the loss weights in Equation (14) and (15). How is the temperature factor set in Equation (2)?
- The output of the encoder should be a 2D feature map. Why use a non-linear head to project it to 256 dimensional vector? Is the vector reshaped to a 2D feature map before fed to the decoder?
- It appears that both the training and the inference are done for 2D slices individually. The 3D continuity information between slices is lost. Such information is important for 3D medical image segmentation, although it might be outside the scope of this paper. It would be better to include a 3D segmentation model for comparison.
- The experiments show that the proposed method is useful when there are few training data (1 to 7 scans). How does the proposed method work when more labeled data are available? It should be studied since they are available in the three datasets. Besides, the training scans are randomly selected from the available labeled data. How does the random selection affect the performance?
- Table 1 does not have the baseline results for reference.




**Main Review:**

The major strength of this paper is the soundness of empirical methodology used to validate the empirical approach. Authors showed intensive experiments and persuasive results to demonstrate the advance of the proposed methods compared with prior works (table 2 and table 3).
The idea of this paper is also with great clinical impact: right now, fully supervised segmentation methods although show good performance, depends heavily on the expensive annotations. It is great to explore the self-/semi- supervised learning algorithms to reduce such strict requirement on data acquisition.

In this paper, authors discussed about the previous related work about self-supervision, contrastive learning and self-paced learning, and later clearly discussed how the proposed method was built upon the prior work.

This work is reproducible. They used public datasets to train and evaluate the algorithm, and the authors promised to make the codes public available if the paper got accepted.

Paper Strength:
+ A new self-paced strategy is proposed for contrastive learning by assigning importance weights to positive sample pairs. With self-paced strategy, the model learns from easy samples in the beginning and gradually add noisy/hard samples.
+ Meta-labels are incorporated in the contrastive learning process, which is reasonable and effective. Using all three meta-labels achieves the best performance.
+ Experiments show that the proposed method outperforms several existing methods on 3D medical image segmentation in the semi-supervised setting. Ablation experiments also demonstrate the effectiveness of the proposed self-paced learning strategy and the use of meta-label in contrastive learning.


**Time Spent Reviewing:**

4 hours

---

> ### Author Response · Authors · 2021-08-11
> **Answer to Reviewer HQGh**
>
> Thank you for your time and insightful feedback!
>
> We will incorporate the changes into the revised paper, and address the issues in detail here:
>
> _Weakness is in the experiment design_:
>
> Our aim was to report the performance of our approach compared to some baselines and show its stability over different datasets with different characteristics. Still, we agree with the reviewer that it would make sense to use only one dataset to select the final method. However, in our experiments, the best results are always obtained with “SP-Con (both) + Mean Teacher”, so no matter the dataset chosen for selecting the final model, we would always select the same best configuration.
>
> _Results on other organs datasets_:
>
> The proposed method could work with any segmentation task where meta-labels are available. This includes segmenting volumetric data of any organ for which a rough correspondence can be obtained between 2D slices in the volume. We have added two new datasets from sub-tasks of Medical Segmentation Decathlon to test the capability of our proposed method to generalize to other organs. These two tasks focused on segmenting the hippocampus and spleen from MRI and CT images, respectively. Due to time limits, we only tested our proposed variant “SP-Con (Pretrain)” on the meta-label “partition” and compared it against strong concurrent approaches such as Contrastive and Mean Teacher.
>
>
> | Hippocampus       | 1 scan | 2 scans | 4 scans |
> |-------------------|--------|---------|---------|
> | baseline          | 60.87  | 73.33   | 78.82   |
> | Mean Teacher      | 70.06  | 75.65   | 80.60   |
> | Contrast          | 64.40  | 75.00   | 81.45   |
> | SP-Con (Pretrain) | 66.70  | 76.89   | 81.25   |
>
> | Spleen            | 2 scans | 4 scans |
> |-------------------|---------|---------|
> | baseline          |  65.51  |  68.59  |
> | Mean Teacher      |  55.02  |  68.10  |
> | Contrast          |  65.14  |  67.21  |
> | SP-Con (Pretrain) |  69.05  |  69.64  |
>
>
> _Algorithmic workflow_:
>
> We created this link to explain the algorithm in pseudo-code:
> https://anonymous.4open.science/r/jlkjdakfhdaskfhdsakfjdasfjdsalkfjsalkdfj/algorithm-flow.png. It will be included in the main paper or in the supplementary material, depending on the space available.
>
> _Metrics not clear_:
>
> In order to clarify our terminology, we will change DSC for DICE score in the paper, especially in the caption of the tables.
> More graphics in the main body: Thank you for the suggestion. Depending on the space available we will include this figure (https://anonymous.4open.science/r/jlkjdakfhdaskfhdsakfjdasfjdsalkfjsalkdfj/diagram_new.png) either in the main paper or in the supplementary material.
>
>
> _Hyper-parameters not specified_:
>
> Indeed, we have some hyperparameters which are essential for the success of our proposed method. The temperature $\tau$ was set to 0.07, close to related works [1,2,3] and works fine across different datasets. For $\lambda $ in Equ. 14, we used $\lambda_1 = 1$, $\lambda_2=1e-3$ and $\lambda_3= 1e-2$  for the ACDC dataset, while $\lambda_1 = 1 $, $\lambda_2=1e-1$ for prostate and MMWHS dataset.  As for our combined method (Equ. 15) which involves a supervised loss, a regularization loss (Mean Teacher) and our SP-based contrastive loss, we fixed $\lambda_{reg}$ as 0.1 and $\lambda_{sp}$ as 0.005 for the ACDC dataset, while $\lambda_{reg}$ being 0.2 and $\lambda_{sp}$ as 0.02 for both the Prostate and MMWHS dataset. We will add this information in the Experimental details section of the Suppl. materials.
>
> [1]: Chen, Ting, et al. "A simple framework for contrastive learning of visual representations." International conference on machine learning. PMLR, 2020.
>
> [2]: Wang, Feng, and Huaping Liu. "Understanding the behaviour of contrastive loss." Proceedings of the IEEE/CVF Conference on Computer Vision and Pattern Recognition. 2021.
>
> [3]: Chaitanya, Krishna, et al. "Contrastive learning of global and local features for medical image segmentation with limited annotations." arXiv preprint arXiv:2006.10511 (2020).
>
>
> _Why use a non-linear head as autoencoder output_:
>
> Following the common practice [1], we added a projector equipped with adaptive average pooling and MLP to project the feature maps to 1-D vectors. This projection is used to reduce the dimensionality of the representation before computing the contrastive loss. Note that the projected representation is not fed to the decoder since we freeze the decoder in pre-training. After the pre-training, we feed the non-projected 2D feature maps, not the projected representation, to the decoder.
>
> _Include a 3D segmentation model for comparison_:
>
> Thanks for the suggestion. We followed the experimental setup of Chaitanya et al. [4] where a 2D segmentation network was employed. However, we would like to mention that 2D-based networks often work better for data having anisotropic acquisition resolutions (z-axis has a large distance compared with x- and y- axes). In particular, 3D networks perform poorly for the ACDC dataset where 2D slices have a large spacing and can be misaligned due to cardiac movement during acquisition [5]. Additionally, 2D networks have a lower computational cost and require less GPU memory than their 3D counterparts.
>
> [4] Chaitanya, Krishna, et al. "Contrastive learning of global and local features for medical image segmentation with limited annotations." arXiv preprint arXiv:2006.10511 (2020).
>
> [5] Baumgartner, Christian F., et al. "An exploration of 2D and 3D deep learning techniques for cardiac MR image segmentation." International Workshop on Statistical Atlases and Computational Models of the Heart. Springer, Cham, 2017.
>
>
> _Performance of the method on entire datasets_:
>
> In the following table, we report the gap between our SP-Con (Pretrain) with slide partition as meta-label versus a Baseline that uses only labelled samples. The values are 3d Dice scores on the ACDC test set.
>
>
> |                          | 1 scan | 2 scans | 4 scans | 8 scans | 175 scans |
> |--------------------------|--------|---------|---------|---------|-----------|
> | Baseline                 | 57.53  | 67.06   | 75.64   | 82.64   | 88.06     |
> | Mean Teacher             | 62.85  | 72.84   | 79.12   | 84.35   | N/A*      |
> | SP-Con (Pretrain)        | 73.99  | 81.01   | 82.83   | 84.29   | 88.35     |
> | Our Improvement          | 16.46  | 13.95   | 7.18    | 1.65    | 0.29      |
>
> *: Mean Teacher requires unlabeled examples for its consistency loss, thus was not considered for the full supervision setting.
>
> We observe that the improvement brought by our method with respect to the baseline (last row in the table) diminishes with the increase of the number of labelled samples. This is expected because when using the entire labelled data (175 scans), there is no additional unlabelled data to exploit. We will add this experiment in the Supplementary Material of this manuscript.
>
> _How does the random selection affect the performance_:
>
> We have recomputed some experiments on the ACDC dataset using different random splits. However, due to time limits, we focused only on our proposed variant SP-Con (Pretrain). We have pretrained the network using self-paced contrastive loss and then fine-tuned it on different splits of labeled data with various meta-labels. We further enforced a fixed random seed which kept the network initialization, data iteration ordering, image augmentation applied on each iteration to be the same across all experiments so that the resulting performance only reflects the difference on data split. We then compared our method with contrastive loss (Chaitanya et al.), Unsupervised contrastive (SimCLR) and Mean Teacher.
>
> The splits are listed as the following:
>
> split1: "patient100_00", "patient027_01".
>
> split2: "patient002_00", "patient017_01".
>
> split3: "patient036_01", "patient076_00".
>
>
> From the reported table we see that different splits result in different performances especially when using a few labelled data. However, we observe that our SP-Con (Pretrain) with L1 outperforms all comparable approaches in all but one case.
>
>
> |                        | split1 |         | split2 |         | split3 |         | mean   |         |
> |------------------------|:------:|---------|--------|---------|:------:|---------|--------|---------|
> |                        | 1 scan | 2 scans | 1 scan | 2 scans | 1 scan | 2 scans | 1 scan | 2 scans |
> | Baseline               |  61.36 | 71.11   |  35.76 | 71.48   |  39.74 | 76.44   |  45.62 | 73.01   |
> | Unsup_Con              |  65.70 | 77.84   |  50.69 | 72.87   |  59.65 | 80.22   |  58.68 | 76.98   |
> | Unsup_Con + SP         |  69.34 | 77.49   |  61.96 | 73.11   |  66.55 | 80.82   |  65.95 | 77.14   |
> | Contrast + L1          |  72.56 | 80.58   |  67.68 | 73.97   |  68.64 | 82.35   |  69.63 | 78.97   |
> | Contrast + L2          |  68.19 | 76.69   |  43.69 | 71.57   |  57.34 | 80.31   |  56.41 | 76.19   |
> | Contrast + L3          |  65.40 | 76.33   |  64.93 | 69.10   |  58.82 | 80.06   |  63.05 | 75.16   |
> | Mean Teacher           |  73.04 | 78.97   |  60.91 | 72.82   |  55.07 | 80.26   |  63.01 | 77.35   |
> | SP-con (pretrain) + L1 |  76.24 | 79.96   |  68.18 | 76.46   |  74.18 | 82.46   |  72.87 | 79.63   |
> | SP-con (pretrain) + L2 |  71.77 | 80.08   |  58.07 | 71.95   |  58.02 | 81.07   |  62.62 | 77.70   |
> | SP-con (pretrain) + L3 |  66.77 | 77.10   |  63.65 | 73.41   |  62.03 | 82.11   |  64.15 | 77.54   |
>
>
> _Table 1 does not have the baseline results for reference_:
>
>  Baseline results for table 1 are the same as in table 2. We will also add them in table 1.

---

> ### Author Response · Authors · 2021-08-27
> **Thanks again for the comments and suggestions**
>
> Dear reviewer HQGh,
>
> We hope our answers have addressed all your concerns in order to upgrade your score.
>
> Please let us know if you have any other questions or comments.

---

> ### Author Response · Authors · 2021-09-02
> **End of discussion phase**
>
> We kindly remind the reviewer that today is the last day to post feedback and modify the score. We hope our answers have addressed all remaining concerns

---

### Decision · Program_Chairs · 2021-09-27

**Decision:**

Accept (Poster)

**Comment:**

The reviewers broadly agreed that the overall motivation for the work (reducing the dependency on labels in medical image segmentation) and the novelty and thorough evaluation of the proposed method are of general interest to the community. There were a few minor clarity points brought up that should be addressed in the revised version, and the authors should also include the additional experimental results from the discussion.